# MeCeFO: Enhancing LLM Training Robustness via Fault-Tolerant Optimization

**Rizhen Hu**[*]
Peking University
rzhu25@stu.pku.edu.cn

**Yutong He**[*]
Peking University
yutonghe@pku.edu.cn

**Ran Yan**
HKUST
ryanaf@connect.ust.hk

**Mou Sun**
Zhejiang Lab
123sssmmm@gmail.com

**Binhang Yuan**[†]
HKUST
biyuan@ust.hk

**Kun Yuan**[†]
Peking University
kunyuan@pku.edu.cn

## Abstract

As distributed optimization scales to meet the demands of Large Language Model (LLM) training, hardware failures become increasingly non-negligible. Existing fault-tolerant training methods often introduce significant computational or memory overhead, demanding additional resources. To address this challenge, we propose **Me**mory- and **C**omputation- **e**fficient **F**ault-tolerant **O**ptimization (**MeCeFO**), a novel algorithm that ensures robust training with minimal overhead. When a computing node fails, MeCeFO seamlessly transfers its training task to a neighboring node while employing memory- and computation-efficient algorithmic optimizations to minimize the extra workload imposed on the neighboring node handling both tasks. MeCeFO leverages three key algorithmic designs: (i) Skip-connection, which drops the multi-head attention (MHA) module during backpropagation for memory- and computation-efficient approximation; (ii) Recomputation, which reduces activation memory in feedforward networks (FFNs); and (iii) Low-rank gradient approximation, enabling efficient estimation of FFN weight matrix gradients. Theoretically, MeCeFO matches the convergence rate of conventional distributed training, with a rate of $\mathcal{O}(1/\sqrt{nT})$, where $n$ is the data parallelism size and $T$ is the number of iterations. Empirically, MeCeFO maintains robust performance under high failure rates, incurring only a 4.18% drop in throughput, demonstrating $5.0\times$ to $6.7\times$ greater resilience than previous SOTA approaches. Codes are available at https://github.com/pkumelon/MeCeFO.

## 1 Introduction

Large language models (LLMs) have demonstrated remarkable capabilities across diverse domains, including machine translation, reasoning, planning, coding, *etc.*, driving their widespread adoption. According to the Chinchilla scaling law [22], model performance scales with the number of model parameters, training tokens, and iterations, necessitating larger model architectures, longer training durations, and, crucially, massive distributed compute resources. For example, Meta's LLaMA 3 405B [17] was trained on 16,000 H100 GPUs for 54 days. Training clusters are continually scaling up, with leading frontier AI efforts now approaching over 100,000 GPUs. At this scale, hardware failures become inevitable—Alibaba reports a downtime percentage of 31.19% for handling failures [12], and Meta reports a frequency of 4 hours per failure on average due to confirmed hardware issues [17]. The

---

[*]Equal contribution.

[†]Corresponding author. Kun Yuan is also affiliated with AI for Science Institute, Beijing, China.

39th Conference on Neural Information Processing Systems (NeurIPS 2025).

potential hardware failures lead to critical challenges for distributed training, degrading GPU utility and training throughput. Such failures present critical challenges for distributed training, reducing GPU utilization and training throughput. The frequency of failures tends to increase with cluster size, making them especially problematic in large-scale systems. This has led to the rise of robust training, which employs fault-tolerant techniques to mitigate the impact of hardware failures.

Existing fault-tolerant approaches in large-scale training focus on *system optimizations*, including checkpointing [15, 40, 36, 56, 63, 2], rescheduling [2, 25], and redundant computing [50]. Checkpointing methods periodically save training states to allow recovery from the most recent checkpoint after a failure. However, beyond the additional memory and computation overhead, replacing failed devices with spares and reloading checkpoints is time-consuming—posing a significant challenge in large-scale training clusters, where failure frequency is high. Rescheduling techniques aim to avoid recovery delays by dynamically reassigning training tasks based on available resources. Nevertheless, a reduced pool of devices still leads to degraded throughput. Redundant computing improves robustness by replicating tasks across multiple devices, but this significantly lowers effective GPU utilization, even when no failures occur. Overall, existing fault-tolerant methods suffer from inefficiencies, as the redundancies required for robustness can substantially degrade training throughput.

It is important to note that the fault-tolerant approaches discussed above are fundamentally *algorithm-agnostic*; that is, their primary goal is to faithfully execute a given training algorithm step-by-step, irrespective of any encountered failures. However, we contend that the ultimate objective of model training is not necessarily to replicate an exact sequence of computations but rather to obtain model parameters that generalize effectively on the intended tasks. Consequently, intermediate results—and even the final outcomes—need not strictly align with those of a fault-free execution scenario. Instead, the critical factor is the performance of the trained model itself. Notably, optimization methods such as stochastic gradient descent (SGD) and Adam [27] inherently exhibit robustness to variations and noise in gradient computations, further supporting this viewpoint. This observation implies that rigid adherence to the original training trajectory might be overly restrictive. Relaxing this constraint could potentially enhance the overall efficiency of fault-tolerant training. This motivates a key question:

*Can we design fault-tolerant optimization algorithms that are more memory- and compute-efficient by strategically sacrificing computation precision, while still achieving strong model performance?*

In response to this question, we propose MeCeFO, a fault-tolerant optimization algorithm for training transformer-based LLMs that reduces the overhead of fault tolerance through memory- and computation-efficient strategies. Specifically, MeCeFO adopts a neighbor-do-both (NDB) strategy, in which a failed node's training task is handled by a neighboring node, which is then responsible for executing both its own task and the failed node's. To alleviate the additional memory overhead on the neighbor node, we introduce a skip-connection technique for the multi-head attention (MHA) module and a recomputation strategy for the feedforward network (FFN). The associated computation overhead is addressed through the combination of skip-connections and a low-rank gradient approximation technique, which compensates for the extra cost introduced by recomputation. Theoretically, we establish a convergence rate of $\mathcal{O}(1/\sqrt{nT})$ for MeCeFO, matching that of standard distributed stochastic gradient descent (SGD). Empirically, our experiments demonstrate that MeCeFO incurs only a 4.18% drop in throughput while maintaining comparable model performance when pre-training LLaMA-7B under high-frequency failures—achieving $5.0\times$ to $6.7\times$ greater resilience than previous state-of-the-art methods. Our contributions include:

- We propose MeCeFO, a novel fault-tolerant optimization algorithm with improved efficiency.

- We theoretically prove that MeCeFO achieves a convergence rate of $\mathcal{O}(1/\sqrt{nT})$ under mild assumptions, matching that of standard distributed SGD.

- We empirically validate MeCeFO across various settings, demonstrating its ability to sustain high training throughput and strong model performance even under frequent failures. In particular, when pre-training LLaMA-7B under high-frequency failure scenarios, MeCeFO incurs only a 4.18% drop in throughput—achieving $5.0\times$ to $6.7\times$ greater resilience than previous state-of-the-art methods.

## 2   Preliminaries and related Works

**Transformer models.** This paper primarily focuses on decoder-only transformer models [53], which are widely adopted in modern large language model designs, including LLaMA [51, 52, 17], GPT [42, 43, 6], DeepSeek [4, 30, 31], *etc.* In general, transformer models consist of multiple transformer blocks, each containing a multi-head attention (MHA) layer followed by a feedforward network (FFN), with both layers equipped with normalization and residual connections [20]. A typical MHA module includes four weight matrices: $W_{\mathrm{q}}$, $W_{\mathrm{k}}$, $W_{\mathrm{v}}$, and $W_{\mathrm{o}}$. The FFN module is usually a shallow MLP; for example, in LLaMA models, the FFN comprises three weight matrices: $W_{\mathrm{gate}}$, $W_{\mathrm{up}}$, and $W_{\mathrm{down}}$. Popular choices of normalization include LayerNorm [3] and RMSNorm [61]. Positional information is typically encoded using positional encodings such as RoPE [48].

**Hybrid parallelism.** There are several parallel computing patterns used in efficient distributed training, such as data parallelism (DP), pipeline parallelism (PP), tensor parallelism (TP), and sequence parallelism (SP), among others. In this work, we focus primarily on the hybrid parallelism setting that combines data parallelism and pipeline parallelism—a popular strategy for training large-scale deep learning models [16, 37, 38, 47]. Specifically, devices are first grouped into different DP ranks, each responsible for processing a different subset of the training data. Within each DP rank, the devices are further organized into a pipeline to handle different layers of the model.

**Memory consumption.** The memory footprint during training consists of four primary components: (i) model weights, (ii) weight gradients, (iii) optimizer states, and (iv) activations. To maximize hardware utilization and training throughput, practitioners often select large mini-batch sizes, which can cause activations to dominate the overall memory consumption, making it a critical bottleneck in large-scale training, particularly for deep networks with high-dimensional intermediate features.

**Computation consumption.** The majority of neural network training computation occurs in dense matrix multiplications within linear layers. Each linear transformation $y = Wx$ involves three key operations: (i) forward propagation (`Fprop`), (ii) weight gradient computation (`Wgrad`), and (iii) activation gradient computation (`Dgrad`) during backpropagation. These operations typically require equivalent amount of computation, resulting in a 1:2 ratio of forward to backward pass computation.

**Fault-tolerant algorithms.** Most existing approaches ensure training robustness through check-pointing. For example, [40] restarts from checkpoints when adjusting resource configurations; [15] resumes training from quantized checkpoints; [36] reduces checkpointing overhead by algorithmically tuning the checkpoint frequency and leveraging pipelining; [56] accelerates checkpointing using NVMe optimizations and write parallelism; [63] restarts from the last saved checkpoint in response to hardware failures or loss divergences; and [2] introduces job morphing to dynamically reconfigure training jobs after restarting from checkpoints with the remaining resources. To avoid the recovery overhead of checkpointing, researchers have also proposed fault-tolerant approaches that do not rely on it. Bamboo [50] employs redundant computation to ensure information availability during failures, while Oobleck [25] precomputes pipeline templates and dynamically adjusts them in response to failures. To the best of our knowledge, this work proposes the first fault-tolerant optimization algorithm that integrates memory- and computation-efficient training techniques to improve efficiency.

**Efficient training.** As memory consumption becomes a major bottleneck in training large-scale models, researchers have developed training algorithms that reduce memory usage. Adapter-based methods such as [23, 41] fine-tune only parameter-efficient additional modules. LoRA [24] and its variants [32, 62, 34, 28] reparameterize dense weight matrices using low-rank adapters to reduce memory costs. [39] proposes randomly activating different layers during training, while [19] combines low-rank and sparse structures for further memory savings. GaLore [64] and its variants [21, 8, 45, 65] apply low-rank gradient approximations to reduce the memory footprint of optimizer states. [26, 60, 35] compress activations to save memory. Most memory-efficient training methods can also enhance training throughput by enabling larger batch sizes [65]. Another line of work improves throughput by directly reducing the number of floating-point operations (FLOPs). For example, DropBP [57] randomly skips connections during backpropagation, while [1, 33, 7] reduce computational cost through approximate matrix multiplication. A major drawback of these memory- and computation-efficient training algorithms is that the resulting models often exhibit a noticeable performance gap compared to standard training. In contrast, MeCeFO applies efficiency techniques only locally and

selectively—specifically when handling failures—allowing it to benefit from efficiency gains without compromising model performance.

## 3   Method

Current fault-tolerant methods are suboptimal for deep learning as they prioritize exact step-by-step computation—a requirement inherited from general distributed computing that is unnecessarily stringent for model training. Unlike rigorous distributed computing tasks, deep learning operates within the framework of distributed stochastic optimization:

$$\min_{\boldsymbol{w} \in \mathbb{R}^d} \quad f(\boldsymbol{w}) := \frac{1}{n} \sum_{i=1}^{n} f_i(\boldsymbol{w}), \quad \text{where } f_i(\boldsymbol{w}) := \mathbb{E}_{\xi \sim \mathcal{D}_i}[F(\boldsymbol{w}; \xi)].$$

Here, $\boldsymbol{w}$ collects all the weight parameters in the model, $n$ denotes the number of data-parallel (DP) ranks, $\mathcal{D}_i$ the local data distribution at the $i$-th rank, and $F$ the per-sample loss function. The key insight is that the training objective targets expected loss minimization rather than exact intermediate computations. Crucially: (i) **Optimizer robustness:** First-order stochastic optimizers (e.g., SGD, Adam) are inherently tolerant to gradient noise; (ii) **Path independence:** The convergence depends on the quality of final weights, not the precise trajectory. This reveals fundamental redundancy in maintaining exact gradient information for training robustness. Leveraging these observations, MeCeFO strategically relaxes exact computation requirements and incorporates memory- and computation-efficient mechanisms to achieve: (i) reduced memory overhead during fault recovery; (ii) lower computational redundancy; and (iii) maintained convergence guarantees.

### 3.1   MeCeFO overview

**Neighbor-do-both strategy.**  Upon detecting a hardware failure, MeCeFO initiates an efficient failover protocol in which the neighbor node within the same DP rank assumes responsibility for both its original workload and the failed node's computational tasks. In the following, we use *failed node* and *neighbor node* to refer to the node occurring hardware failures and the node that takes charge of the failed node's workload, respectively. Although the failed node's memory (including model weights and optimizer states) becomes inaccessible to the rest of the training network, this information is not entirely lost due to the inherent memory redundancy in data parallelism, which maintains identical backups across other DP ranks. In MeCeFO, the neighbor node directly retrieves the required information—including the failed node's model weights and optimizer states——from the corresponding device responsible for the same layers in another DP group.

However, naively implementing the neighbor-do-both (NDB) mechanism significantly degrades training efficiency. The neighbor node must maintain doubled memory footprint and computational load during failures, creating two key bottlenecks: (i) **Memory inefficiency**: Each GPU must reserve half of its total memory capacity to accommodate the additional layers during failure scenarios. This not only reduces memory utilization but also forces smaller macro-batch sizes to avoid out-of-memory (OOM) errors, directly impacting computational throughput; (ii) **Pipeline imbalance**: Processing doubled computational loads increases execution time proportionally. This creates pipeline bubbles that propagate within the data parallel (DP) rank, forcing other devices to remain idle while waiting for the overloaded node to complete its computations.

To address these challenges, we develop specialized computation- and memory-efficient techniques. Our solution incorporates three key innovations: (i) **Skip-connection:** We reduce both computational load (Wgrad and Dgrad) and activation memory requirements in the MHA modules through strategic skip-connections; (ii) **Selective activation recomputation:** For feed-forward network (FFN) modules, we implement an efficient recomputation strategy that maintains only critical activation checkpoints, dramatically reducing memory demands; (iii) **Low-rank gradient approximation:** We introduce a novel approximation technique for FFN weight gradients that significantly decreases computational complexity of the Wgrad operations, effectively compensating for the overhead introduced by recomputation. Alg. 1 provides a general view of the proposed MeCeFO algorithm.

**Remark.** MeCeFO is designed as a complementary component within broader fault-tolerant frameworks, contributing to the construction of more resilient large-scale training systems. Rather than endorsing a specific solution, we highlight representative examples to illustrate the feasibility and

---

**Algorithm 1** MeCeFO Algorithm

---

**Input:** Initial model weights $\boldsymbol{w}^{(0)} = \{\boldsymbol{W}_{\ell,\#}\}_{\#\in\{q,k,v,o,gate,up,down\}}^{1\leq\ell\leq L}$, projection matrix update frequency $\tau$.

**Output:** Final model weights $\boldsymbol{w}^{(T)}$.

 1: Initialize NDB steps $t_{i,\ell} \leftarrow 0$ for rank $i$, layer $\ell$;
 2: **for** global step $t = 0$ to $T - 1$ **do**
 3:     **for** DP ranks $i = 1$ to $n$ **do in parallel**
 4:         Check node availability and rearrange tasks according to the NDB strategy;
 5:         **for** layer $\ell$ in failed nodes **do**             ▷ on failure
 6:             Neighbor node fetches $\boldsymbol{W}_{\ell}^{(t)}$ and optimizer states from other DP ranks;
 7:             Reset local step $t_{i,\ell} \leftarrow 0$;
 8:         **end for**
 9:         **for** layer $\ell$ in recovered nodes **do**         ▷ on recovery
10:             Original node fetches $\boldsymbol{W}_{\ell}^{(t)}$ and optimizer states from the neighbor node;
11:         **end for**
12:         **for** layer $\ell = 1$ to $L$ **do**
13:             Execute forward pass `MeCeFO_Forward`$(\boldsymbol{W}_{\ell}^{(t)})$;     ▷ forward pass
14:         **end for**
15:         **for** layer $\ell = L$ to $1$ **do**
16:             Execute backward pass `MeCeFO_Backward`$(\boldsymbol{W}_{\ell}^{(t)}, t_{i,\ell})$;    ▷ backward pass
17:             Get averaged gradients $\overline{\boldsymbol{G}}_{\ell}^{(t)}$ according to (1);    ▷ gradient averaging
18:             Execute optimizer step to get $\boldsymbol{W}_{\ell}^{(t+1)}$ according to gradient $\overline{\boldsymbol{G}}_{\ell}^{(t)}$;   ▷ optimizer step
19:         **end for**
20:     **end for**
21: **end for**

---

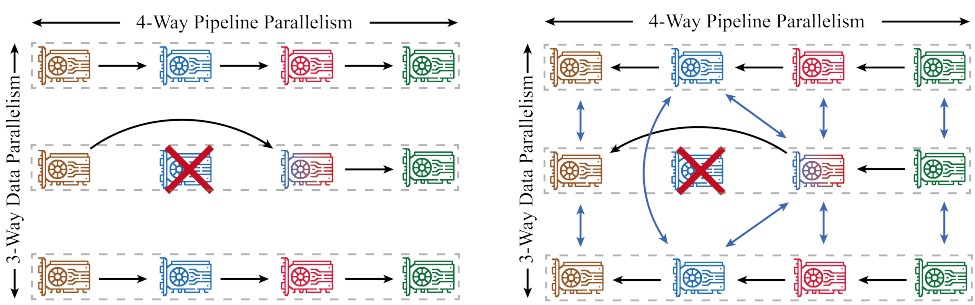

Figure 1: Overview of the MeCeFO framework. During both forward (left) and backward (right) propagation, the workload of a failed node is offloaded to a neighboring node within the same data parallel (DP) group.

flexibility of integration: MeCeFO effectively addresses isolated node failures; [18] has proposed hierarchical detection mechanisms that target switch-level and interconnect failures in distributed environments; [58] has considered broader challenges, such as node freezing, communication disruptions, and software errors. By combining MeCeFO with such system-level techniques, one can construct a more comprehensive and reliable fault-tolerant infrastructure for distributed training.

## 3.2 Key technique I: skip-connection

MeCeFO's skip-connection technique draws inspiration from DropBP [57]. While DropBP randomly skips connections in both the MHA and FFN modules with varying probabilities, MeCeFO employs a deterministic strategy that consistently skips the MHA module connections and maintains connectivity for the FFN module, as illustrated in Fig. 2. This design choice stems from our empirical observation (Fig. 3) that skipping MHA connections introduces significantly less training disruption than alternative choices. When neighbor nodes skip MHA in the backward pass, gradient contributions come

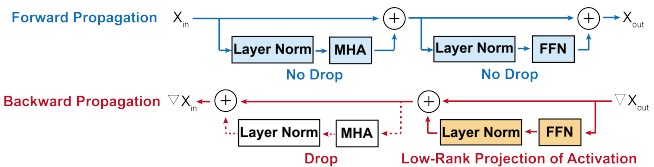

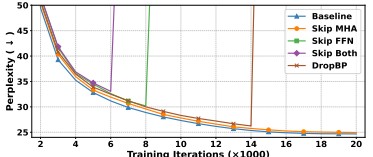

Figure 2: The skip-connection technique in MHA layers. We only skip the MHA's connection in the backward propagation.

Figure 3: Ablation of module skipping in LLaMA-130M pre-training.

exclusively from unaffected DP ranks. Formally, the averaged gradients are computed as:

$$\overline{\boldsymbol{G}}_{\ell,\#} = \frac{1}{|\mathcal{N}_{\ell,\#}|} \sum_{i \in \mathcal{N}_{\ell,\#}} \boldsymbol{G}_{i,\ell,\#}, \tag{1}$$

where $\# \in \{\mathrm{q}, \mathrm{k}, \mathrm{v}, \mathrm{o}\}$, $\mathcal{N}_{\ell,\#} \subseteq \{1, 2, \cdots, n\}$ represents active DP ranks where the device responsible for training weight matrix $\boldsymbol{W}_{\ell,\#}$ in the $\ell$-th layer is neither failed nor serving as a neighbor node of a failed one, and $\boldsymbol{G}_{i,\ell,\#}$ represents the stochastic gradient regarding $\boldsymbol{W}_{\ell,\#}$ computed by DP $i$. When all DP ranks are unaffected, *i.e.*, $\mathcal{N}_{\ell,\#} = \{1, 2, \cdots, n\}$, (1) reduces to the standard format:

$$\overline{\boldsymbol{G}}_{\ell,\#} = \frac{1}{n} \sum_{i=1}^{n} \boldsymbol{G}_{i,\ell,\#}.$$

**Memory efficiency.** The skip-connection technique eliminates the need for neighbor nodes to store activations in MHA modules, significantly reducing memory overhead when handling doubled tasks.

**Computation efficiency.** The skip-connection design eliminates the need for neighbor nodes to compute `Wgrad` and `Dgrad` in MHA modules, significantly reducing the computation costs.

### 3.3 Key technique II: selective activation recomputation

Unlike MHA modules, applying skip-connections to FFN modules proves problematic for two key reasons: (i) FFN-skip-connections introduce substantial approximation errors in input activation gradients, severely degrading backpropagation quality; (ii) In failure-prone scenarios, reduced participation of DP ranks for FFN weight updates exacerbates data heterogeneity issues, leading to non-negligible gradient bias. To maintain training stability while preserving memory efficiency, we instead employ activation recomputation for FFN modules. Specifically, neighbor nodes only maintain the input to each FFN modules and recompute all other activations during back propagation.

**Memory efficiency.** The recomputation technique eliminates the need for neighbor nodes to store intermediate activations in FFN modules, significantly reducing the memory overhead.

**Computation overhead.** The recomputation technique introduces additional computational costs for neighbor nodes. Specifically, each FFN module requires one additional forward pass (`Rcomp`). This recomputation cost is equivalent to a standard `Fprop` operation. The total overhead amounts to approximately one third of the baseline FFN computation cost in normal training scenarios.

### 3.4 Key technique III: low-rank gradient approximation

Although the memory cost for both the MHA module and the FFN module has been significantly reduced by techniques I and II, only MHA's computation cost has been reduced, and FFN's computation cost has been increased by 1/3. To mitigate this issue, we propose the following low-rank gradient approximation technique to compensate for the recomputation overhead. Specifically, for a linear layer $\boldsymbol{y} = \boldsymbol{W}\boldsymbol{x}$ in the FFN module with $\boldsymbol{W} \in \mathbb{R}^{m \times n}$, we conduct singular value decomposition (SVD) of $\boldsymbol{W}$ yielding $\boldsymbol{W} = \boldsymbol{U}\boldsymbol{\Sigma}\boldsymbol{V}^{\top}$. Let $\boldsymbol{V}_1 = \boldsymbol{V}[:, :r]$ collect the top-$r$ right singular vectors—first $r$ columns of $\boldsymbol{V}$—we have the following approximation:

$$\boldsymbol{G}_{\mathrm{W}} = \boldsymbol{G}_{\mathrm{y}}\boldsymbol{x}^{\top} = \boldsymbol{G}_{\mathrm{y}}\boldsymbol{x}^{\top}\boldsymbol{V}\boldsymbol{V}^{\top} \approx \boldsymbol{G}_{\mathrm{y}}(\boldsymbol{x}^{\top}\boldsymbol{V}_1)\boldsymbol{V}_1^{\top}. \tag{2}$$

Here, $\boldsymbol{G}_{\mathrm{y}}$ represents the gradient of activation $\boldsymbol{y}$. We only recompute matrix $\boldsymbol{V}_1$ every $\tau$ iteration to further reduce the SVD overhead.

---

**Algorithm 2** MeCeFO Forward Pass

---

**def** `MeCeFO_Forward`$(\boldsymbol{W}_\ell^{(t)})$:
1: **if** node not taking doubled workload **then**          ▷ standard node
2:   Execute standard MHA and FFN forward pass with all activations maintained;
3: **else**                    ▷ neighbor node
4:   Execute standard MHA and FFN with only input activations to FFN maintained;
5: **end if**

---

**Algorithm 3** MeCeFO Backward Pass

---

**def** `MeCeFO_Backward`$(\boldsymbol{W}_\ell^{(t)}, t_{i,\ell})$:
1: **if** node no taking doubled workload **then**         ▷ standard node
2:   Execute standard MHA and FFN backward pass yielding gradients $\boldsymbol{G}_{i,\ell}^{(t)}$;
3: **else**                    ▷ neighbor node
4:   **if** $t_{i,\ell} \equiv 0 \ (\mathrm{mod}\ \tau)$ **then**      ▷ compute projection matrix every $\tau$ iterations
5:    $\boldsymbol{W}_{\ell,\#}^{(t)} = \boldsymbol{U}_{\ell,\#}^{(t)} \boldsymbol{\Sigma}_{\ell,\#}^{(t)} \boldsymbol{V}_{\ell,\#}^{(t)}$, $\tilde{\boldsymbol{V}}_{i,\ell,\#}^{(t)} \leftarrow \boldsymbol{V}_{\ell,\#}^{(t)}[:,:r]$, $\# \in \{\mathrm{gate}, \mathrm{up}, \mathrm{down}\}$;
6:   **else**
7:    $\tilde{\boldsymbol{V}}_{i,\ell,\#}^{(t)} \leftarrow \tilde{\boldsymbol{V}}_{i,\ell,\#}^{(t-1)}$, $\# \in \{\mathrm{gate}, \mathrm{up}, \mathrm{down}\}$;        ▷ reuse projection
8:   Recompute FFN activations using the maintained input activations;
9:   Apply (2) to approximate $\boldsymbol{G}_{i,\ell,\#}^{(t)}$ via $\tilde{\boldsymbol{V}}_{i,\ell,\#}^{(t)}$, $\# \in \{\mathrm{gate}, \mathrm{up}, \mathrm{down}\}$;
10:   Skip the MHA connection and propagate activation gradients to previous layers;
11:   Update local step $t_{i,\ell} \leftarrow t_{i,\ell} + 1$;
12:   **end if**
13: **end if**

---

**Memory overhead.** The additional memory for $\boldsymbol{V}_1 \in \mathbb{R}^{n \times r}$ is negligible when $r \ll \min\{m, n\}$.

**Computation efficiency.** Let $b$ denote the batch size times sequence length. Compared with the original FLOPs $2bmn$ to compute $\boldsymbol{G}_\mathrm{W} = \boldsymbol{G}_\mathrm{y} \boldsymbol{x}^\top$, applying the low-rank gradient approximation technique requires only $(2brn + 2brm + 2rmn)$ FLOPs. When $r \ll \min\{b, m, n\}$, the approximated `Wgrad` operation is computationally negligible, approximately compensating for the `Rcomp` overhead.

## 4 Convergence analysis

**Assumption 1** (Lower-boundedness and $L$-smoothness). *We assume function* $f : \mathbb{R}^d \to \mathbb{R}$ *satisfies*

$$\inf_{\boldsymbol{w} \in \mathbb{R}^d} f(\boldsymbol{w}) > -\infty, \quad and \quad \|\nabla f(\boldsymbol{w}) - \nabla f(\boldsymbol{w}')\|_2 \le L\|\boldsymbol{w} - \boldsymbol{w}'\|_2, \ \forall \ \boldsymbol{w}, \boldsymbol{w}' \in \mathbb{R}^d.$$

**Assumption 2** (Stochastic gradient). *We assume the stochastic gradient oracles satisfy*

$$\mathbb{E}_{\xi \sim \mathcal{D}_i}[\nabla F(\boldsymbol{w}; \xi)] = \nabla f_i(\boldsymbol{w}), \quad and \quad \mathbb{E}_{\xi \sim \mathcal{D}_i}[\|\nabla F(\boldsymbol{w}; \xi) - \nabla f_i(\boldsymbol{w})\|_2^2] \le \sigma^2,$$

*for* $\forall \ i \in \{1, 2, \cdots, n\}$ *and some* $\sigma > 0$.

**Assumption 3** (Gradient error). *We assume that the following inequalities hold for MeCeFO's approximated gradient* $\overline{\boldsymbol{g}}^{(t)}$ *during the optimization process* $t = 0, 1, \cdots, T - 1$:

$$\left\| \overline{\boldsymbol{g}}^{(t)} - \overline{\boldsymbol{g}}_\star^{(t)} \right\|_2^2 \le (1 - \delta) \left\| \overline{\boldsymbol{g}}_\star^{(t)} \right\|_2^2,$$

$$\left\| \mathbb{E}_{\xi_i \sim \mathcal{D}_i} \left[ \overline{\boldsymbol{g}}^{(t)} \right] - \nabla f(\boldsymbol{w}^{(t)}) \right\|_2^2 \le (1 - \delta) \left\| \nabla f(\boldsymbol{w}^{(t)}) \right\|_2^2,$$

*where* $\delta \in (0, 1]$ *and* $\overline{\boldsymbol{g}}_\star^{(t)}$ *represents the fault-free averaged stochastic gradient at step* $t$.

**Remark.** Assumptions 1 and 2 are standard assumptions commonly used in convergence analysis. To validate Assumption 3, we empirically observe the relative errors through our experiments. Specifically, we observe the single-batch relative error $\|\overline{\boldsymbol{g}}^{(t)} - \overline{\boldsymbol{g}}_\star^{(t)}\|_2^2 / \|\overline{\boldsymbol{g}}_\star^{(t)}\|_2^2$ and full-batch relative

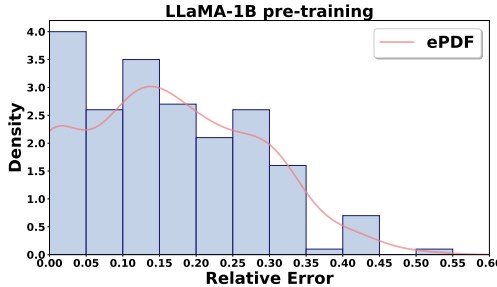
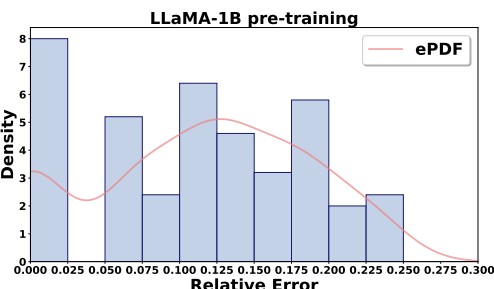

Figure 4: Single-batch relative error of pre-training LLaMA-1B on the C4 dataset.

Figure 5: Full-batch relative error of pre-training LLaMA-1B on the C4 dataset.

Table 1: Configuration of Different Failure Scenarios

| Scenario Name | Failure Interval | Node Recovery Time |
|---|---|---|
| Low Frequency Failure | Every 2 hours | Every 4 hours |
| Medium Frequency Failure | Every 1 hour | Every 3 hours |
| High Frequency Failure | Every 0.5 hours | Every 2 hours |

error $\|\mathbb{E}_{\xi_i \sim \mathcal{D}_i}[\overline{\boldsymbol{g}}^{(t)}] - \nabla f(\boldsymbol{w}^{(t)})\|_2^2 / \|\nabla f(\boldsymbol{w}^{(t)})\|_2^2$ while pre-training LLaMA-1B. As illustrated in Fig. 4 and 5, these errors are consistently smaller than 0.6, justifying the application of Assumption 3.

Below we present the convergence results of MeCeFO.

**Theorem 1.** *Under Assumptions 1-3, if momentum parameter $\beta_1 \in (1-\delta/(24-12\delta), 1)$ and learning rate $\eta \leq \min\{1/(2L), \sqrt{(\delta(1-\beta_1)^2)/(8L^2)}\}$, MeCeFO (with momentum SGD) converges as*

$$\frac{1}{T+1}\sum_{t=0}^{T}\mathbb{E}[\|\nabla f(\boldsymbol{w}^{(t)})\|_2^2] \leq \frac{8\Delta}{\delta\eta(T+1)} + \frac{8\Delta_1}{\delta(1-\beta_1)(T+1)} + \frac{24(1-\beta_1)\sigma^2}{\delta n},$$

*where $\Delta := f(\boldsymbol{w}^{(0)}) - \inf_{\boldsymbol{w}} f(\boldsymbol{w})$, and $\Delta_1 := \|\boldsymbol{m}^{(0)} - \nabla f(\boldsymbol{w}^{(0)})\|_2^2$. (Proof is in Appendix A)*

**Corollary 1.** *Under Assumptions 1-3, if we choose $\eta = \left(2L + \sqrt{\frac{8L^2}{\delta(1-\beta_1)^2}}\right)^{-1}$ and $\beta_1 = 1 - \left(\frac{24}{\delta} + \sqrt{\frac{\delta^{1/2}(T+1)\sigma^2}{n(L\Delta+\delta\Delta_1)}}\right)^{-1}$, MeCeFO (with momentum SGD) converges as*

$$\frac{1}{T+1}\sum_{t=0}^{T}\mathbb{E}[\|\nabla f(\boldsymbol{w}^{(t)})\|_2^2] = \mathcal{O}\left(\sqrt{\frac{(L\Delta+\delta\Delta_1)\sigma^2}{\delta^{5/2}n(T+1)}} + \frac{L\Delta+\delta\Delta_1}{\delta^{5/2}(T+1)}\right),$$

*matching standard distributed SGD's convergence rate of $\mathcal{O}\left(\frac{\sigma}{\sqrt{nT}} + \frac{1}{T}\right)$.*

## 5 Experimental results

In this section, we empirically evaluate MeCeFO across various scenarios to assess its training performance. Ablation results and additional details are provided in Appendix C and D.

### 5.1 Experimental setup

**Cluster setup.** We conducted experiments on a 32-GPU cluster composed of four nodes, each with eight NVIDIA A100 GPUs. Intra-node communication leveraged NVLink (600 GB/s), and inter-node communication used InfiniBand (200 GB/s).

**Baselines.** In our experiments, we compare MeCeFO against two state-of-the-art fault-tolerant training methods, Bamboo [50] and Oobleck [25], using their publicly available implementations.

**Workloads.** We pre-train LLaMA [51] models of various sizes on the C4 [44] dataset, using different global batch sizes and training iterations for each configuration. Specifically,

Table 2: Throughput Performance and Degradation under Different Fault Frequencies

| Model | System | Throughput (tokens/s) | | | | Throughput Drop (%) | | |
|---|---|---|---|---|---|---|---|---|
| | | No Fault | Low Freq. | Mid Freq. | High Freq. | Low Freq. | Mid Freq. | High Freq. |
| LLaMA-350M | Bamboo | 438.06k | 428.90k | 421.45k | 407.22k | 2.09 | 3.79 | 7.04 |
| | Oobleck | 703.73k | 674.15k | 662.93k | 632.40k | 4.20 | 5.80 | 10.14 |
| | **MeCeFO** | **1199.23k** | **1197.39k** | **1193.25k** | **1186.35k** | **0.15** | **0.50** | **1.07** |
| LLaMA-1B | Bamboo | 153.75k | 146.91k | 144.66k | 141.13k | 4.45 | 5.91 | 8.21 |
| | Oobleck | 291.05k | 276.05k | 268.29k | 250.68k | 5.16 | 7.82 | 13.87 |
| | **MeCeFO** | **471.19k** | **464.79k** | **461.23k** | **457.13k** | **1.36** | **2.11** | **2.98** |
| LLaMA-7B | Bamboo | 12.41k | 11.45k | 10.74k | 9.82k | 7.73 | 13.42 | 20.84 |
| | Oobleck | 66.95k | 57.05k | 51.63k | 48.14k | 14.78 | 22.87 | 28.09 |
| | **MeCeFO** | **111.12k** | **108.15k** | **107.70k** | **106.47k** | **2.67** | **3.08** | **4.18** |

Table 3: Validation Perplexities of LLaMA Models Pre-trained by MeCeFO under Different Fault Frequencies

| Model | No Fault | Low-frequency Fault | Medium-frequency Fault | High-frequency Fault |
|---|---|---|---|---|
| LLaMA-350M | 18.74 | 18.75 | 18.88 | 19.04 |
| LLaMA-1B | 15.49 | 15.51 | 15.61 | 15.83 |
| LLaMA-7B | 14.92 | 14.97 | 15.04 | 15.16 |

- **LLaMA-350M:** Trained for 6,000 iterations with a global batch size of 8,192.

- **LLaMA-1B:** Trained for 20,000 iterations with a global batch size of 4,096.

- **LLaMA-7B:** Trained for 60,000 iterations with a global batch size of 1,024.

**Failure Scenario.** We simulate three distinct failure scenarios, each defined by a specific failure frequency and recovery time of corresponding nodes. The detailed configurations are summarized in Table 1. These scenarios impose varying levels of stress on the system, from stable (low-frequency) to moderately disrupted (medium-frequency), and highly volatile (high-frequency). This design allows us to evaluate the system's fault tolerance and recovery behavior under different failure intensities.

## 5.2 Training throughput under failures

Comparisons across frameworks under fault-free conditions and varying fault frequencies highlight the strong performance of MeCeFO. The throughput of each method is summarized in Table 2.

In the fault-free setting, MeCeFO maintains a throughput of 1,199.23k tokens/s with LLaMA-350M, dropping only slightly to 1,186k tokens/s under high-frequency faults—a mere 1.07% degradation. Similar robustness is observed for LLaMA-1B (2.98% degradation) and LLaMA-7B (4.18%).

In contrast, Bamboo experiences a $2.76\times$ to $13.9\times$ throughput drop compared to MeCeFO across different settings. Due to its reliance on redundant computation, Bamboo also suffers from low throughput even under fault-free conditions. For instance, in LLaMA-350M pre-training, it achieves only 438.06k tokens/s—substantially lower than both Oobleck (703.73k tokens/s) and MeCeFO (1199.23k tokens/s). As a result, while Bamboo's relative throughput degradation under faults may appear modest, its heavy resource overhead fundamentally limits overall performance.

Oobleck, focused on system-level optimizations, exhibits significant throughput degradation as fault frequency increases, ranging from $3.71\times$ to $28.0\times$ worse than MeCeFO. For LLaMA-350M, the degradation reaches 10.14% under high-frequency faults and escalates to 28.09% for LLaMA-7B.

These results support our perspective that strictly adhering to conventional optimization algorithms in fault-tolerant training can be unnecessarily restrictive. By relaxing this constraint and incorporating memory- and computation-efficient learning techniques, it is possible to significantly enhance training efficiency, highlighting that efficient fault-tolerant design goes beyond purely system-level solutions.

## 5.3 Training performance under failures

To evaluate MeCeFO's impact on training convergence, we measured the validation perplexity of LLaMA-350M, LLaMA-1B, and LLaMA-7B trained with MeCeFO under different failure scenarios.

Table 4: Zero-shot evaluation scores of LLaMA-1B Pre-trained by MeCeFO under Different Fault Frequencies

| Fault Frequencies | BoolQ[9] | ARC-Easy [10] | PIQA [5] | TruthfulQA-MC2 [29] | Avg. |
|---|---|---|---|---|---|
| No Fault | 0.579 | **0.459** | 0.682 | 0.427 | 0.537 |
| Low Freq. | **0.594** | 0.455 | 0.674 | **0.451** | **0.544** |
| Mid Freq. | 0.571 | 0.446 | 0.678 | 0.425 | 0.530 |
| High Freq. | 0.587 | 0.454 | **0.684** | 0.417 | 0.536 |

Table 5: Fine-tuning Results on Pre-trained LLaMA-1B under Corresponding Fault Frequencies

| Fault Frequencies | CoLA | STS-B | MRPC | RTE | SST2 | MNLI | QNLI | QQP | Avg. |
|---|---|---|---|---|---|---|---|---|---|
| No Fault | 46.93 | **89.21** | **89.12** | 62.61 | **92.36** | **81.82** | 88.61 | 89.83 | 80.06 |
| Low Freq. | 46.86 | 89.14 | 88.92 | 62.59 | 92.31 | 81.78 | 88.58 | **90.07** | 80.03 |
| Mid Freq. | **47.21** | 89.14 | 88.84 | **63.18** | 92.25 | 81.80 | 88.61 | 90.02 | **80.13** |
| High Freq. | 46.67 | 89.16 | 88.87 | 62.58 | 92.30 | 81.71 | **88.66** | 89.94 | 79.99 |

To further assess downstream capabilities, we evaluated LLaMA-1B models pre-trained with MeCeFO on several zero-shot tasks and conducted fine-tuning experiments on the GLUE [54] benchmark under corresponding failure scenarios.

**Pre-training performance.** As shown in Table 3, the increase in perplexity caused by MeCeFO's efficient training strategies under failure conditions is minimal. Under high-frequency faults, the perplexity for LLaMA-350M increases slightly from 18.74 to 19.04 (1.60%); for LLaMA-1B, from 15.49 to 15.83 (2.19%); and for LLaMA-7B, from 14.92 to 15.16 (1.61%). Under medium- and low-frequency fault scenarios, the increases are even smaller—less than 0.80% and 0.34%, respectively.

**Zero-shot performance.** As shown in Table 4, the pre-trained LLaMA-1B models maintain robust zero-shot performance across all failure scenarios. Compared to the fault-free baseline (0.537 average), the average scores are 0.544 under low-frequency faults, 0.530 under mid-frequency faults, and 0.536 under high-frequency faults. Notably, under low-frequency faults, MeCeFO even yields slight improvements on BoolQ (0.594 vs. 0.579) and TruthfulQA-MC2 (0.451 vs. 0.427), leading to the highest overall average. These results demonstrate that MeCeFO preserves, and in some cases enhances, the downstream generalization ability of the model despite frequent failures.

**Fine-tuning performance.** As shown in Table 5, LLaMA-1B models pre-trained with MeCeFO under different failure scenarios achieve downstream performance on GLUE that is nearly identical to the fault-free baseline. The average score of the baseline model (80.06) is well preserved: 80.03 under low-frequency faults, 80.13 under mid-frequency faults, and 79.99 under high-frequency faults. In particular, the mid-frequency fault model slightly surpasses the baseline on CoLA (47.21 vs. 46.93) and RTE (63.18 vs. 62.61), leading to the highest overall average.

These findings confirm that MeCeFO effectively maintains training performance. Its ability to sustain comparable perplexity metrics even under high-frequency fault conditions demonstrates robust fault tolerance without significant compromise to final model quality.

# 6 Conclusions and limitations

We propose MeCeFO, a fault-tolerant training algorithm that achieves high efficiency through three core techniques: (i) skip-connection, (ii) selective activation recomputation, and (iii) low-rank gradient approximation. Theoretically, MeCeFO retains a convergence rate of $\mathcal{O}(1/\sqrt{nT})$, matching that of standard distributed SGD. Empirically, MeCeFO incurs only a 4.18% throughput degradation when pre-training LLaMA-7B under high-frequency failures while maintaining comparable model performance. In contrast, existing SOTA methods that strictly adhere to fault-free assumptions suffer $5.0\times$ to $6.7\times$ greater throughput degradation. Our study has several limitations, including the use of a per-iteration failure setting, limited access to large-scale fault-prone clusters for experiments, and the reliance of our theoretical results on Assumption 3, which we plan to address in future work.

## Acknowledgments and Disclosure of Funding

This work is supported by the National Key Research and Development Program of China (No. 2024YFA1012902) and National Natural Science Foundation of China (No. 124B2017, 92370121, 12301392, W2441021).

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

# A Proof of Theorem 1

First, we specify the update rules of MeCeFO with momentum SGD as follows:

$$\boldsymbol{m}^{(t)} = \beta_1 \boldsymbol{m}^{(t-1)} + (1 - \beta_1)\overline{\boldsymbol{g}}^{(t)},$$
$$\boldsymbol{w}^{(t+1)} = \boldsymbol{w}^{(t)} - \eta \boldsymbol{m}^{(t)},$$

where $\overline{g}^{(t)}$ is MeCeFO's averaged weight gradient, $\boldsymbol{m}^{(-1)} = \boldsymbol{0}$, $\beta_1 \in (0,1)$ is the momentum parameter, $\eta > 0$ is the learning rate.

Next, we present several key lemmas.

**Lemma 1** (Descent lemma). *Under Assumption 1, it holds that*

$$f(\boldsymbol{w}^{(t+1)}) \leq f(\boldsymbol{w}^{(t)}) - \left(\frac{1}{2\eta} - \frac{L}{2}\right)\|\boldsymbol{w}^{(t+1)} - \boldsymbol{w}^{(t)}\|_2^2 + \frac{\eta}{2}\|\nabla f(\boldsymbol{w}^{(t)}) - \boldsymbol{m}^{(t)}\|_2^2$$
$$- \frac{\eta}{2}\|\nabla f(\boldsymbol{w}^{(t)})\|_2^2. \tag{3}$$

*Proof of Lemma 1.* By $L$-smoothness of $f$ (Assumption 1), we have

$$f(\boldsymbol{w}^{(t+1)}) \leq f(\boldsymbol{w}^{(t)}) + \langle \nabla f(\boldsymbol{w}^{(t)}), \boldsymbol{w}^{(t+1)} - \boldsymbol{w}^{(t)} \rangle + \frac{L}{2}\|\boldsymbol{w}^{(t+1)} - \boldsymbol{w}^{(t)}\|_2^2. \tag{4}$$

For the inner product, we have

$$\langle \nabla f(\boldsymbol{w}^{(t)}), \boldsymbol{w}^{(t+1)} - \boldsymbol{w}^{(t)} \rangle$$
$$= \left\langle \frac{\boldsymbol{m}^{(t)}}{2}, \boldsymbol{w}^{(t+1)} - \boldsymbol{w}^{(t)} \right\rangle + \left\langle \nabla f(\boldsymbol{w}^{(t)}) - \frac{\boldsymbol{m}^{(t)}}{2}, \boldsymbol{w}^{(t+1)} - \boldsymbol{w}^{(t)} \right\rangle$$
$$= -\frac{1}{2\eta}\|\boldsymbol{w}^{(t+1)} - \boldsymbol{w}^{(t)}\|_2^2 + \frac{\eta}{2}\|\nabla f(\boldsymbol{w}^{(t)}) - \boldsymbol{m}^{(t)}\|_2^2 - \frac{\eta}{2}\|\nabla f(\boldsymbol{w}^{(t)})\|_2^2, \tag{5}$$

where the last equality uses $\boldsymbol{w}^{(t+1)} - \boldsymbol{w}^{(t)} = -\eta \boldsymbol{m}^{(t)}$. Applying (5) to (4) yields (3). $\qquad\square$

**Lemma 2** (Momentum-gradient gap). *Under Assumptions 1, 2 and 3, it holds that*

$$\frac{1}{T+1}\sum_{t=0}^{T}\mathbb{E}[\|\boldsymbol{m}^{(t)} - \nabla f(\boldsymbol{w}^{(t)})]$$
$$\leq \frac{2\Delta_1}{(1-\beta_1)(T+1)} + \frac{4L^2}{\delta(1-\beta_1)^2} \cdot \frac{1}{T+1}\sum_{t=1}^{T}\mathbb{E}[\|\boldsymbol{w}^{(t)} - \boldsymbol{w}^{(t-1)}\|_2^2]$$
$$+ \left(1 - \frac{\delta}{2}\right)(7 - 6\beta_1) \cdot \frac{1}{T+1}\sum_{t=1}^{T}\mathbb{E}[\|\nabla f(\boldsymbol{w}^{(t)})\|_2^2] + \frac{6(1-\beta_1)\sigma^2}{n}, \tag{6}$$

*where* $\Delta_1 := \|\boldsymbol{m}^{(0)} - \nabla f(\boldsymbol{w}^{(0)})\|_2^2$.

*Proof of Lemma 2.* According to the update rules, we have

$$\mathbb{E}[\|\boldsymbol{m}^{(t)} - \nabla f(\boldsymbol{w}^{(t)})\|_2^2] = \mathbb{E}[\|\beta_1(\boldsymbol{m}^{(t-1)} - \nabla f(\boldsymbol{w}^{(t)})) + (1-\beta_1)(\overline{\boldsymbol{g}}^{(t)} - \nabla f(\boldsymbol{w}^{(t)}))\|_2^2]$$
$$= \mathbb{E}[\|\beta_1(\boldsymbol{m}^{(t-1)} - \nabla f(\boldsymbol{w}^{(t)})) + (1-\beta_1)(\mathbb{E}[\overline{\boldsymbol{g}}^{(t)}] - \nabla f(\boldsymbol{w}^{(t)}))\|_2^2$$
$$+ (1-\beta_1)^2\mathbb{E}[\|\overline{\boldsymbol{g}}^{(t)} - \mathbb{E}[\overline{\boldsymbol{g}}^{(t)}]\|_2^2]. \tag{7}$$

For the first term, applying Jensen's inequality yields

$$\mathbb{E}[\|\beta_1(\boldsymbol{m}^{(t-1)} - \nabla f(\boldsymbol{w}^{(t)})) + (1-\beta_1)(\mathbb{E}[\overline{\boldsymbol{g}}^{(t)}] - \nabla f(\boldsymbol{w}^{(t)}))\|_2^2]$$
$$\leq \beta_1\mathbb{E}[\|\boldsymbol{m}^{(t-1)} - \nabla f(\boldsymbol{w}^{(t)})\|_2^2] + (1-\beta_1)\mathbb{E}[\|\mathbb{E}[\overline{\boldsymbol{g}}^{(t)}] - \nabla f(\boldsymbol{w}^{(t)})\|_2^2]$$
$$\leq \beta_1\mathbb{E}[\|\boldsymbol{m}^{(t-1)} - \nabla f(\boldsymbol{w}^{(t)})\|_2^2] + (1-\delta)(1-\beta_1)\mathbb{E}[\|\nabla f(\boldsymbol{w}^{(t)})\|_2^2], \tag{8}$$

where the last inequality uses Assumption 3. By Young's inequality, we have

$$\mathbb{E}[\|\boldsymbol{m}^{(t-1)} - \nabla f(\boldsymbol{w}^{(t)})\|_2^2] \leq \left(1 + \frac{\delta(1-\beta_1)}{2}\right) \mathbb{E}[\|\boldsymbol{m}^{(t-1)} - \nabla f(\boldsymbol{w}^{(t-1)})\|_2^2]$$
$$+ \left(1 + \frac{2}{\delta(1-\beta_1)}\right) \mathbb{E}[\|\nabla f(\boldsymbol{w}^{(t)}) - \nabla f(\boldsymbol{w}^{(t-1)})\|_2^2]. \quad (9)$$

For the second term, applying Cauchy's inequality yields

$$\mathbb{E}[\|\overline{\boldsymbol{g}}^{(t)} - \mathbb{E}[\overline{\boldsymbol{g}}^{(t)}]\|_2^2]$$
$$\leq 3\mathbb{E}[\|\overline{\boldsymbol{g}}^{(t)} - \overline{\boldsymbol{g}}_\star^{(t)}\|_2^2] + 3\mathbb{E}[\|\overline{\boldsymbol{g}}_\star^{(t)} - \nabla f(\boldsymbol{w}^{(t)})\|_2^2] + 3\mathbb{E}[\|\nabla f(\boldsymbol{w}^{(t)}) - \mathbb{E}[\overline{\boldsymbol{g}}^{(t)}]\|_2^2]$$
$$\leq 3(1-\delta)\mathbb{E}[\|\overline{\boldsymbol{g}}_\star^{(t)}\|_2^2] + \frac{3\sigma^2}{n} + 3(1-\delta)\mathbb{E}[\|\nabla f(\boldsymbol{w}^{(t)})\|_2^2]$$
$$\leq 6(1-\delta)\mathbb{E}[\|\nabla f(\boldsymbol{w}^{(t)})\|_2^2] + \frac{(6-3\delta)\sigma^2}{n}, \quad (10)$$

where the second inequality uses Assumptions 2 and 3, the last inequality uses Assumption 2. Applying (8)(9)(10) to (7) and using Assumption 1, we obtain

$$\mathbb{E}[\|\boldsymbol{m}^{(t)} - \nabla f(\boldsymbol{w}^{(t)})\|_2^2]$$
$$\leq \left(1 - (1-\beta_1)\left(1 - \frac{\delta}{2}\right)\right)\mathbb{E}[\|\boldsymbol{m}^{(t-1)} - \nabla f(\boldsymbol{w}^{(t-1)})\|_2^2] + \frac{2L^2}{\delta(1-\beta_1)}\mathbb{E}[\|\boldsymbol{w}^{(t)} - \boldsymbol{w}^{(t-1)}\|_2^2]$$
$$+ (1-\delta)(1-\beta_1)(7-6\beta_1)\mathbb{E}[\|\nabla f(\boldsymbol{w}^{(t)})\|_2^2] + \frac{(6-3\delta)(1-\beta_1)^2\sigma^2}{n}. \quad (11)$$

Summing (11) from $t = 1$ to $T$ yields (6). $\qquad\square$

Now we are ready to prove Theorem 1. We restate Theorem 1 as follows.

**Theorem 2** (Convergence of MeCeFO)**.** *Under Assumptions 1-3, if $\beta_1 \in (1 - \delta/(24 - 12\delta), 1)$ and $\eta \leq \min\{1/(2L), \sqrt{(\delta(1-\beta_1)^2)/(8L^2)}\}$, MeCeFO (with momentum SGD) converges as*

$$\frac{1}{T+1}\sum_{t=0}^{T}\mathbb{E}[\|\nabla f(\boldsymbol{w}^{(t)})\|_2^2] \leq \frac{8\Delta}{\delta\eta(T+1)} + \frac{8\Delta_1}{\delta(1-\beta_1)(T+1)} + \frac{24(1-\beta_1)\sigma^2}{\delta n}, \quad (12)$$

*where $\Delta := f(\boldsymbol{w}^{(0)}) - \inf_{\boldsymbol{w}} f(\boldsymbol{w})$, and $\Delta_1 := \|\boldsymbol{m}^{(0)} - \nabla f(\boldsymbol{w}^{(0)})\|_2^2$.*

*Proof of Theorem 2.* Summing (3) in Lemma 1 for $t = 0, 1, \cdots, T$ and taking expectation, we have

$$\inf_{\boldsymbol{w}\in\mathbb{R}^d} f(\boldsymbol{w}) - f(\boldsymbol{w}^{(0)}) \leq \frac{\eta}{2}\sum_{t=0}^{T}\mathbb{E}[\|\nabla f(\boldsymbol{w}^{(t)}) - \boldsymbol{m}^{(t)}\|_2^2] - \left(\frac{1}{2\eta} - \frac{L}{2}\right)\sum_{t=0}^{T}\mathbb{E}[\|\boldsymbol{w}^{(t+1)} - \boldsymbol{w}^{(t)}\|_2^2]$$
$$- \frac{\eta}{2}\sum_{t=0}^{T}\mathbb{E}[\|\nabla f(\boldsymbol{w}^{(t)})\|_2^2]. \quad (13)$$

Applying Lemma 2 to (13) and noting that $\beta_1 \in (1 - \delta/(24 - 12\delta), 1)$ implies $(1 - \delta/2)(7 - 6\beta_1) \leq 1 - \delta/4$, we obtain

$$\frac{1}{T+1}\sum_{t=0}^{T}\mathbb{E}[\|\nabla f(\boldsymbol{w}^{(t)})\|_2^2] \leq -\frac{8}{\delta\eta}\left(\frac{1}{2\eta} - \frac{L}{2} - \frac{2\eta L}{\delta(1-\beta_1)^2}\right)\frac{1}{T+1}\sum_{t=0}^{T}\mathbb{E}[\|\boldsymbol{w}^{(t+1)} - \boldsymbol{w}^{(t)}\|_2^2]$$
$$+ \frac{8\Delta}{\delta\eta(T+1)} + \frac{8\Delta_1}{\delta(1-\beta_1)(T+1)} + \frac{24(1-\beta_1)\sigma^2}{\delta n}. \quad (14)$$

Noting that $\eta \leq \min\{1/(2L), \sqrt{(\delta(1-\beta_1)^2)/(8L^2)}\}$ implies $1/(4\eta) \geq L/2$ and $1/(4\eta) \geq (2\eta L^2)/(\delta(1-\beta_1)^2)$, (12) is a direct result of (14). $\qquad\square$

Table 6: Performance and Memory Usage Comparison of Models with Varying Batch Sizes. "OOM" denotes an Out of Memory error. A hyphen (-) indicates data not available.

| Method | Batch Size = 256 | | Batch Size = 512 | |
|---|---|---|---|---|
| | Throughput (tokens/s) | Memory (GB) | Throughput (tokens/s) | Memory (GB) |
| MeCeFOmrl | 19.11k | 76.13 | - | OOM |
| MeCeFOrl | 30.23k | 54.65 | - | OOM |
| MeCeFOl | 26.41k | 38.48 | 23.86k | 70.71 |
| MeCeFO | 28.06k | 39.21 | 27.19k | 73.25 |
| MeCeFO w/o Fault | 28.12k | 41.52 | 30.04k | 76.05 |

## B  Discussions

**Experimental setup of failure scenarios.** Although the experimental setup assumes uniformly random failures, real-world failure patterns are often asymmetric or localized. From a theoretical perspective, MeCeFO remains robust in such settings, since fallback operations continue to balance data exposure across pipelines without introducing systematic bias. To support this claim, we further simulated persistent failures on a fixed subset of GPUs and observed validation perplexities that closely matched those under uniform random failures (see Appendix C.2). In addition, the failure-to-recovery ratios reported in Table 1 highlight the increasing challenges of repair and maintenance in larger computing systems; further discussion on the implications of these ratios can be found in Appendix C.3.

**Extension to other parallel strategies.** While our main discussions focus on the DP+PP setting, the design of MeCeFO is inherently local to each node, as its three core mechanisms—skip connections, selective activation recomputation, and low-rank gradient approximation—are applied independently at the node level. This locality allows MeCeFO to naturally extend to TP scenarios. In the event of a failed node within a TP group, the workload can be redistributed across sibling TP ranks within the same PP stage, avoiding the recomputation of the entire group. When $|\text{TP}| > 2$, the resulting overhead per node is strictly less than $1\times$, which makes it feasible to adopt conservative fallback strategies for better error control. Furthermore, the mechanisms in MeCeFO are tunable: skip connections may be applied to only a subset of sub-modules, gradient checkpointing can retain additional activations to reduce recomputation depth, and low-rank gradient approximation can employ higher ranks for improved fidelity. These adaptations ensure that MeCeFO remains compatible with TP while providing flexibility in balancing efficiency and accuracy.

**Transfer potential to soft-error scenarios.**  In addition to hard-fault tolerance, there exists a complementary line of work on mitigating soft errors and stragglers. For instance, replication and redundancy mechanisms have been proposed to prevent undetected computational errors that may corrupt outputs [13] and to alleviate the performance impact of slow workers in distributed systems [49]. Our method takes a different perspective by tolerating bounded training errors in exchange for reduced computational redundancy, thereby improving efficiency while preserving robustness to hard faults. We view this perspective as complementary to existing approaches, and it may inspire future extensions of MeCeFO toward soft-error resilience or straggler mitigation.

## C  Ablation studies and additional results

### C.1  Ablation on key techniques in MeCeFO

We conducted ablation experiments to assess the contribution of each technique to training efficiency. These experiments were carried out on a server with 8 A100 GPUs using pipeline parallelism to train the LLaMA-7B model. "MeCeFO w/o Fault" denotes baseline training without node failures, while all other setups involved a single node failure during training. "MeCeFO" refers to the full proposed fault-tolerant algorithm. To evaluate individual components, we designed the following variants:

- **MeCeFOmrl**: MeCeFO without skip-connection, selective activation recomputation and low-rank gradient approximation (key techniques I, II and III).

Table 7: Validation perplexities of LLaMA-1B trained with MeCeFO under asymmetric (static subset) vs. symmetric (uniform random) failures.

| Failure Setting | No Fault | Low Freq. | Mid Freq. | High Freq. |
|---|---|---|---|---|
| Asymmetric | 15.49 | 15.54 | 15.62 | 15.75 |
| Symmetric | 15.49 | 15.51 | 15.61 | 15.83 |

Table 8: Configurations and validation perplexities of LLaMA-1B pre-trained with MeCeFO.

| Scenario Name | Failure Interval | Node Recovery Time | Perplexity |
|---|---|---|---|
| High Frequency Failure | Every 30 minutes | Every 120 minutes | 15.83 |
| Higher Frequency Failure | Every 10 minutes | Every 40 minutes | 15.81 |

- **MeCeFOrl**: MeCeFO without selective acivation recomputation and low-rank gradient approximation (key techniques II and III).

- **MeCeFOl**: MeCeFO without low-rank gradient approximation (key technique III).

According to Table 6, removing all techniques (MeCeFOmrl) leads to a sharp increase in memory footprint from 41.52GB to 76.13GB for the neighbor node when resuming training with a batch size of 256. At the same time, throughput drops significantly from 28.12k tokens/s to 19.11k tokens/s. With a batch size of 512, this configuration triggers an OOM (Out of Memory) error. These results indicate that using the NDB strategy alone is impractical.

For the MeCeFOrl variant, an OOM error still occurred at a batch size of 512, indicating that dropping only MHA activations is insufficient to alleviate the memory pressure caused by the doubled workload.

In the MeCeFOl variant, the throughput at a batch size of 256 decreased from 28.12k tokens/s to 26.41k tokens/s. This suggests that although recomputing FFN activations helps reduce memory usage, the added computational overhead negatively impacts throughput.

Finally, the full MeCeFO algorithm, integrating all optimization components, achieved a throughput of 28.06k tokens/s and a memory footprint of 39.21GB under a batch size of 256 in a single-node failure scenario—closely approaching the performance of fault-free training.

These experimental results confirm that each component of the MeCeFO scheme plays a critical role in either memory optimization or computational efficiency. Their synergistic integration enables the system to sustain high throughput and effectively prevent memory overflows, even under fault conditions with reduced computational resources.

## C.2    Ablation on asymmetric failures

We conducted an ablation study simulating persistent non-uniform failures. Specifically, we randomly selected 5 GPUs to fail repeatedly throughout the entire training process, while the remaining GPUs remained fully operational. All other experimental settings were identical to those used in the main study. The validation perplexities are summarized in Table 7, where the asymmetric setting closely matches the symmetric (uniform) failure case.

The results indicate that even in the presence of persistent and localized failures, MeCeFO maintains robustness without significant degradation in training quality.

## C.3    Ablation on failure scenarios

In fact, it is the *ratio* between failure and recovery rates—rather than their absolute values—that is more relevant to training performance under failures, as it determines the steady-state proportion of healthy nodes and thus the overall system behavior and algorithmic robustness. To examine this effect, we pre-trained the LLaMA-1B model with MeCeFO under a new failure scenario named *Higher Frequency Failure* where failures occur every 10 minutes and recoveries take 40 minutes, i.e., both events are more frequent while preserving the same ratio as in the high-frequency setting.

Table 9: Equivalent failure and recovery rates under different scenarios.

| Scenario | Sim. Cluster | Freq. Per GPU Per Hour | | #Real Nodes Per GPU | Freq. Per Real Node Per Hour | |
| --- | --- | --- | --- | --- | --- | --- |
| | | Fail Freq. | Recov. Freq. | | Fail Freq. | Recov. Freq. |
| Low Freq. | 32 GPUs | 1/64 | 1/128 | $N$ | $1/(64N)$ | $1/(128N)$ |
| Mid Freq. | 32 GPUs | 1/32 | 1/96 | $2N$ | $1/(64N)$ | $1/(192N)$ |
| High Freq. | 32 GPUs | 1/16 | 1/64 | $4N$ | $1/(64N)$ | $1/(256N)$ |

Table 10: Validation perplexities of MLA+MoE-1.2B pre-trained with MeCeFO.

| Model | No Fault | Low Freq. | Mid Freq. | High Freq. |
| --- | --- | --- | --- | --- |
| MLA+MoE-1.2B | 16.17 | 16.22 | 16.37 | 16.43 |

The resulting validation perplexity was 15.81, which is nearly identical to the 15.83 obtained in the original high-frequency scenario (see Table 8).

**Remark.** To further align our experimental setup with realistic large-scale deployments, we intentionally amplify failure and recovery events on a 32-GPU cluster to emulate systems with hundreds or even thousands of nodes. In this abstraction, each simulated GPU corresponds to $N \gg 1$ real nodes, each with a much lower per-node failure or recovery rate. As summarized in Table 1, this setup ensures that the equivalent failure frequency per real node remains consistent across different scenarios, while the equivalent recovery frequency per real node decreases, reflecting the increasing difficulty of repair and maintenance in larger-scale clusters. A summary of this mapping between simulated and equivalent real-node clusters is provided in Table 9.

### C.4 Results on other model structures

We further evaluate MeCeFO on a Deepseek V3-style model that integrates Multi-Head Latent Attention (MLA) [31] and Mixture-of-Experts (MoE) [46], with a total of 1.2B parameters and 0.1B active parameters. As shown in Table 10, the validation perplexities of this model trained with MeCeFO remain consistently comparable across all failure scenarios.

### C.5 Results on extended validation of Assumption 3.

Deeper models may exhibit longer error propagation paths. To assess the generality of our assumption beyond the 1B case, we conducted additional experiments on a 7B model with 32 transformer layers, constrained by our current computational resources. The results shown in Fig. 6 and 7 reveal similar trends in gradient approximation error, suggesting that Assumption 3 remains valid at larger scales.

## D Experimental specifications

This section provides detailed descriptions of our experimental setup, covering algorithm implementation, model specifications, and training configurations.

**Implementation.** We implement MeCeFO on top of the HexiScale framework [59], which itself builds upon Megatron-LM [38].

**Parallel strategies.** We use $|\text{DP}| = 4$ and $|\text{PP}| = 8$ throughout all experiments.

**Model specifications.** Table 11 presents the detailed configurations of LLaMA-350M, LLaMA-1B, and LLaMA-7B, including hidden dimensions, FFN intermediate dimensions, number of attention heads, and layers. A maximum sequence length of 256 is used across all experiments.

**Training configurations.** Across all scenarios, we use the AdamW optimizer with $\beta_1 = 0.9$, $\beta_2 = 0.999$, `weight_decay` $= 0.01$, and $\epsilon = 1 \times 10^{-8}$. A learning rate warmup is applied over the first 10% of training iterations, followed by a cosine annealing schedule that decays the learning rate to 10% of its initial value. For MeCeFO, the SVD frequency is set to $\tau = 100$. The number of training steps, batch sizes, and initial learning rates are listed in Table 11 and are tuned exclusively for optimizing baseline fault-free training performance.

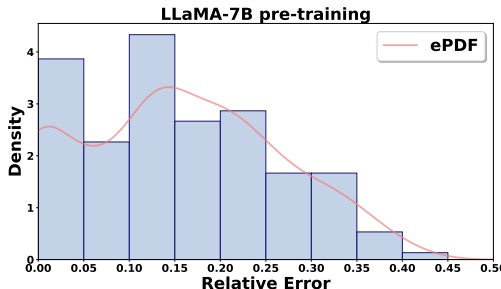

Figure 6: Single-batch relative error of pre-training LLaMA-7B on the C4 dataset.

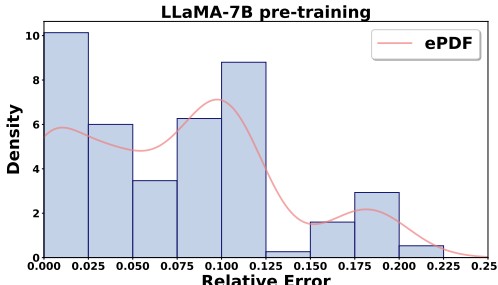

Figure 7: Full-batch relative error of pre-training LLaMA-7B on the C4 dataset.

Table 11: Architecture and Hyperparameters of Different LLaMA Models.

| Model | Hidden | Intermediate | Heads | Layers | Steps | Batch Size | Learning Rate |
|-------|--------|--------------|-------|--------|-------|------------|---------------|
| LLaMA-350M | 1024 | 2736 | 16 | 24 | 6k | 8192 | $8 \times 10^{-4}$ |
| LLaMA-1B | 2048 | 5461 | 32 | 24 | 20k | 4096 | $6 \times 10^{-4}$ |
| LLaMA-7B | 4096 | 11008 | 32 | 32 | 60k | 1024 | $4 \times 10^{-4}$ |

**Failure Modeling in Fault-Tolerant Computing.** Our work is related to fault-tolerant computing and reliability in distributed training. Prior studies have often modeled system failures using exponential or shifted-exponential distributions motivated by straggler effects [11, 55, 14]. In contrast, our focus is primarily on sudden hardware failures (e.g., node crashes), which we approximate as memoryless events. This motivates the adoption of a Poisson process assumption, under which each node is assigned a constant failure probability per iteration. Importantly, if one adopts a stricter time-based Poisson model, then methods with lower throughput (i.e., longer iteration times) would experience higher effective failure probabilities. Since baseline methods tend to suffer more severe throughput degradation under failure than MeCeFO, such a model would further amplify the relative advantage of our approach.

