# OpenReview forum: "MeCeFO: Enhancing LLM Training Robustness via Fault-Tolerant Optimization"
_NeurIPS.cc/2025/Conference — NeurIPS 2025 poster_

### Official Review · Reviewer_Ebnf · 2025-07-02

**Clarity:** 3
**Significance:** 4
**Originality:** 4
**Rating:** 5
**Confidence:** 4

**Summary:**

The paper proposes a new strategy of training LLMs (distributed optimization) in the presence of hardware failures. The strategy is called Memory- and Computation- efficient Fault-tolerant Optimization (MeCeFO) which uses some interesting algorithmic innovations, like skip connections, recomputation, and low-rank gradient approximation. They have provided a theoretical guarantee and also included experimental results on LLAMA 350M, 1B, and 7B, comparing with two baselines: (1) Bamboo (2) Oobleck. The proposed strategy shows improvements over these methods (under simulated node failures).

**Questions:**

Could you highlight which steps of the proof are altered under this scheme? Is it only the approximate gradient step?

Could you highlight if and how the theoretical guarantee depends on the nature of the failure, e.g., if the system is more/less failure-prone, shouldn't it affect the theoretical guarantee?

**Ethical Concerns:**

["NO or VERY MINOR ethics concerns only"]

**Final Justification:**

The authors have addressed my comments. I have increased my rating.

**Limitations:**

One sentence on limitations is included. But, this could be expanded to include some of the points mentioned in Weakness.

-- Other kinds of failure models beyond constant frequency

-- Simulated vs Real failures/delays

-- Other kinds of approaches in fault-tolerant computing that are based on replication/ECCs

-- Theoretical guarantees

**Paper Formatting Concerns:**

N.A.

**Quality:**

3

**Strengths And Weaknesses:**

STRENGTHS

1. This is a very interesting and relevant problem. They have motivated the issue of hardware failures quite well, citing practical challenges faced by Alibaba and Meta.

2. The proposed strategy seems quite interesting, and combines three types of algorithmic innovations, diving deep into the nuances of matrix operations.

3. Experiments are provided over several LLMs for pre-training, which is not an easy feat (also requires significant computational resources). They have compared with two baselines and demonstrate considerable improvements. Ablation studies are included in the Appendix.

WEAKNESSES

1. The theoretical guarantee is much appreciated, but it seems to follow from standard assumptions and convergence guarantees of momentum SGD in distributed settings. Could you highlight which steps of the proof are altered under this scheme? Is it only the approximate gradient step? Could you highlight if and how the theoretical guarantee depends on the nature of the failure, e.g., if the system is more/less failure-prone, shouldn't it affect the theoretical guarantee?

2. It is not very clear if and how the system failure is modelled into the mathematical analysis. Do you assume a particular statistical model for the failure? In the experiments, a constant frequency failure model has been considered.

Prior works in fault-tolerant computing have also considered exponential runtimes, motivated by the straggling effect (causes node delays which could also be an interesting motivational use-case for this paper). Some of these references could be included in the motivation/related works:

[1] Tail at Scale: https://research.google/pubs/the-tail-at-scale/

[2] Using Straggler Replication to Reduce Latency in Large-scale Parallel Computing: https://dl.acm.org/doi/abs/10.1145/2847220.2847223

[3] Short-Dot: Computing Large Linear Transforms Distributedly Using Coded Short Dot Products : https://proceedings.neurips.cc/paper_files/paper/2016/file/aace49c7d80767cffec0e513ae886df0-Paper.pdf

3. I guess the hardware failures are simulated - under a constant frequency failure model. It would be interesting future work to study other kind of failures, e.g., by a Poisson process.

4. Curious if it possible to see any kind of real failures rather than simulated ones, either from delays (straggling) or failures? Could you comment on whether the clusters were dedicatedly set aside during these experiments, or were other jobs also being scheduled on them during this time (potentially leading to natural delays/failures)?

5. Prior works in fault-tolerant computing have studied alternate approaches [4,5] to perform reliable training of machine learning models under stragglers/hardware failures - by using replication or error-correction to have extra computations so that one can proceed even if some nodes fail/straggle. This direction of research would be a relevant complementary approach for the proposed technique [4,5], and could be discussed in related works if not compared with.

[4] CodeNet: Training Large Scale Neural Networks in Presence of Soft-Errors: https://arxiv.org/abs/1903.01042

[5] Gradient Coding: Avoiding Stragglers in Distributed Learning: https://proceedings.mlr.press/v70/tandon17a/tandon17a.pdf

---

> ### Author Rebuttal · Authors · 2025-07-31
>
> We thank the reviewer for acknowledging our method and experimental results, as well as the detailed comments and suggestions. All questions have been clarified as best as we can, and we are glad to address any further comments or questions.
>
> ---
> > **Weakness 1 (Question 1 and 2). Could the authors highlight which steps of the proof are altered under the scheme and how the theoretical guarantees depend on the failure?**
>
> Thank you for the thoughtful question. The main modifications in the theoretical analysis occur in **Lemma 2**, specifically in equations **(8)** and **(10)**, where we apply **Assumption 3** to upper-bound the bias introduced by the approximate gradient.
>
> * If there were **no failure** or **no gradient approximation**, the analysis would reduce to the standard momentum SGD proof (e.g., by setting $\delta = 1$).
> * In our case, $\delta$ captures the **quality of the gradient approximation**, which is **directly affected by the failure rate**. A higher failure rate leads to **a larger proportion of approximated gradients**, which results in a **smaller value of $\delta$** under Assumption 3.
>
> This dependency is reflected in **Corollary 1**, where the convergence rate is explicitly influenced by $\delta$: the **smaller the $\delta$**, the **slower the convergence**. Thus, the **impact of failures on theoretical convergence** is clearly accounted for through this parameter.
>
> ---
> > **Weakness 2. It is not clear how the system failure is modeled into mathematical analysis. Prior works [1-3] in fault-tolerant computing have considered exponential runtimes motivated by the straggling effect, which could be included in the motivation/related works.**
>
> Thank you for the valuable references. For mathematical analysis, the failure is modelled through $\delta$ in Assumption 3. For experimental design, due to limited computational resources, we did not collect real-world failure traces to fit a detailed statistical model. Our focus in this work is primarily on **sudden hardware failures**, such as node crashes, rather than **latency or straggler effects**.
>
> We assume that such hardware failures are approximately **memoryless**, which motivates our use of a **Poisson process** as an underlying assumption. Given that iteration times are relatively stable under our setup, this further justifies modeling each node with a **constant failure probability** per iteration.
>
> Moreover, we assign the same failure/recovery probability to each node for two main reasons:
>
> 1. The failure simulation is intended to estimate the behavior of large-scale training under uniform stress, rather than to capture heterogeneous failure behavior.
> 2. Under the **i.i.d. data distribution** assumption, the specific identity of the failed node (i.e., which data-parallel worker it belongs to) does not substantially alter the training dynamics.
>
> We agree that more refined models like **shifted exponential distributions**, as used in [2], are highly relevant and could enrich our framework, especially in extending MeCeFO to cover straggler-style delays. We thank the reviewer for suggesting this direction and will incorporate these discussions and references into the revised version of the paper.
>
> ---
> > **Weakness 3. It would be interesting future work to study other kind of failures, e.g., by a Poisson process.**
>
> Thank you for the insightful comment. As we mentioned earlier, if the iteration time is approximately constant—which is the case in our experiments due to stable training throughput—then our **constant per-iteration failure probability** is essentially equivalent to a **Poisson process** in continuous time.
>
> If one were to adopt a stricter Poisson process model where failures are measured per unit time rather than per iteration, then methods with **lower throughput (longer iteration time)** would experience **higher failure probabilities**. The baseline methods suffer from **more severe throughput degradation** compared to MeCeFO under failure, this would effectively make their performance worse under a true Poisson process. In contrast, MeCeFO would benefit relatively more, potentially **amplifying its advantage** in such a setting.
>
> We appreciate the reviewer for raising this interesting point, which we believe strengthens the case for MeCeFO under realistic failure conditions. We will include this discussion in the revised version of the paper.
>
> ---
> > **Weakness 4. Curious if it possible to see any kind of real failures rather than simulated ones. Whether the clusters were dedicatedly set aside during these experiments, or were other jobs also being scheduled on them during this time?**
>
> Thank you for the question. Due to hardware resource limitations, our study primarily focuses on **simulated failures** rather than real hardware failures or delays, following a common setup in the literature [6]. While we recognize the importance of evaluating real-world failure scenarios (e.g., stragglers or transient faults), it is challenging to observe and control such events reliably without access to large-scale dedicated infrastructure.
>
> To ensure the **validity of our throughput measurements**, we ran all experiments on **dedicated nodes**, without any other concurrent jobs scheduled on the same machines. This allowed us to isolate the effects of simulated failures and ensure consistency across runs.
>
> We agree that incorporating real system-level failures or straggling behaviors would be a valuable direction for future work.
>
> > [6] Jang, I., Yang, Z., Zhang, Z., Jin, X., & Chowdhury, M. (2023, October). Oobleck: Resilient distributed training of large models using pipeline templates. In Proceedings of the 29th Symposium on Operating Systems Principles (pp. 382-395).
>
> ---
> > **Weakness 5. Prior works have studied alternate approaches [4,5] to perform reliable training of machine learning models by using replication or error-correction to have extra computations. This direction of research would be a relevant complementary approach and could be discussed in related works.**
>
> We thank the reviewer for the valuable suggestion. We plan to add the following discussion to the paper:
>
> > In addition to addressing **hard errors** such as node dropouts or hardware failures, there has been substantial work on **soft-error tolerance** in distributed training. For example, [4] focuses on preventing undetected errors that may cause nodes to produce garbage outputs, while [5] addresses the straggler problem by using redundancy to mitigate slow workers in distributed setups.
> > We believe that the core idea of our work—**tolerating a bounded amount of training error in exchange for reduced computational redundancy**—offers a novel perspective for improving robustness to hard faults. This approach may also inspire future research into complementary domains such as soft-error resilience or straggler mitigation.
>
> ---
> > **Limitation. The limitation could be expanded to include some of the points mentioned in Weakness.**
>
> We thank the reviewer for the constructive suggestion. In the revised version, we will expand the discussion of limitations to explicitly include the following points:
>
> - **Failure modeling**: Our current model assumes a constant per-iteration failure probability, which aligns with a Poisson process under the assumption of stable throughput. However, it does not capture more complex or dynamic failure patterns such as straggling delays or correlated failures.
> - **Simulated vs. real failures**: Due to limited hardware access, our experiments simulate failure events rather than observing real hardware failures or delays. Investigating MeCeFO’s behavior under real-world system failures would be an important future direction.
> - **Complementarity with redundancy-based approaches**: Our method is orthogonal to prior fault-tolerant strategies based on replication or error-correcting codes. While those methods aim to recover exact results through redundancy, our approach tolerates a small amount of error to preserve throughput, making it particularly suitable for large-scale or LLM training tasks.
> - **Theoretical guarantees**: Our analysis relies on Assumption 3 (bounded deviation of failure-influenced gradients), which may not always hold in more complex or adversarial settings. Extending the theory to relax this assumption is also an important future direction.
>
> We will include these points explicitly in the limitations section of the paper.
>
> ---
> We thank the reviewer again for the careful comments and valuable suggestions. We hope these responses can clarify the reviewer's questions and are more than happy to address any further comments or questions.

---

> > ### Comment · Reviewer_Ebnf · 2025-08-05
> > **Read the response**
> >
> > Thank you for your detailed response! I have updated my score.
> > The edits suggested here should be included in the final version of the paper.

---

> > > ### Author Response · Authors · 2025-08-07
> > >
> > > Dear Reviewer Ebnf,
> > >
> > > Thank you for your positive feedback and for taking the time to review our work. We truly appreciate your support and are glad that our clarifications addressed your concerns.
> > >
> > > Best regards,
> > >
> > > Authors of Submission 15836

---

### Official Review · Reviewer_16Gg · 2025-07-03

**Clarity:** 4
**Significance:** 4
**Originality:** 4
**Rating:** 6
**Confidence:** 4

**Summary:**

This paper proposed a novel training algorithm MeCoFO designed to improve the fault tolerance of LLM training while minimizing additional memory and computation overhead.

**Questions:**

1. One claim is that MeCeFO avoids having to reserve half the GPU memory for potential failures. Can the authors provide more insight or data on memory usage?
2. The experiments simulate up to a high frequency failure scenario (a failure every 30 minutes, with 2-hour recovery). If failures were even more frequent or recovery slower, how would that affect training?
3. It would be great to see MeCeFO applied to other training regimes, such as fine-tuning or different model types, just to confirm its broad utility.

**Ethical Concerns:**

["NO or VERY MINOR ethics concerns only"]

**Final Justification:**

Thank you for your detailed response.

It has resolved all my concerns, and I have decided to maintain my rating score, and increase my confidence score to 4.

**Limitations:**

yes

**Paper Formatting Concerns:**

The submission appears to be well-formatted according to the NeurIPS guidelines.

**Quality:**

4

**Strengths And Weaknesses:**

Strengths:
1. This paper targets the important problem of hardware failures in large-scale LLM training.
2. MeCoFO shows impressively low overhead.
3. Instead of traditionally recovering through checkpoint etc. MeCeFO utilized the noise robust nature of SGD.
4. The author gives theoretically proof that MeCeFO achieves a convergence rate of $O(1/\sqrt{nT})$ under mild assumptions, demonstrating the robustness of MeCeFO algorithm.

Weakness:
1. The author mentioned that "MeCeFO is primarily tailored for specific hybrid parallel strategies and transformer-based architectures" which leads to a concern if this algorithm functions on other parallel strategies.
2. They also note that they haven't explores full system-level optimizations in combination with their method. I appreciate this transparency, and don't see any glaring additional limitations.

---

> ### Author Rebuttal · Authors · 2025-07-31
>
> We thank the reviewer for acknowledging our theoretical and experimental results, as well as the detailed comments and suggestions. All questions have been clarified as best as we can, and we are glad to address any further comments or questions.
>
> ---
> > **Weakness 1. MeCeFO is tailored for specific hybrid parallel strategies. Could the authors discuss if this algorithm functions on other parallel strategies?**
>
> We appreciate the reviewer’s concern regarding the generality of MeCeFO under different parallel strategies. As noted in our Limitations section, our current implementation and evaluation are focused on a specific hybrid parallel setup commonly used in large-scale transformer training.
>
> That said, **MeCeFO is fundamentally designed to reduce per-node computational and memory overhead**, thereby allowing surviving nodes to take on additional workload during failure recovery. This core principle is **independent of the particular parallel strategy** and remains valid under other setups, including tensor parallelism (TP).
>
> While we have not implemented MeCeFO under TP settings, we believe the extension is natural. Specifically, the three key techniques—**Skip Connection Rescheduling**, **Selective Gradient Checkpointing**, and **Adaptive Low-Rank Approximation**—are all applied locally within a node and can be adjusted to reflect the level of workload rebalancing required by different parallel granularities.
>
> For example:
>
> - Under TP, when a failed node must be covered by sibling TP ranks in the same pipeline stage, the increase in per-node overhead is generally moderate (e.g., <2× when TP > 2). MeCeFO can thus employ more **conservative configurations** of the three techniques to ensure minimal impact on accuracy.
> - The ratio of skipped submodules in **Skip Connection Rescheduling**, the extent of recomputation in **Selective Gradient Checkpointing**, and the target rank in **Adaptive Low-Rank Approximation** can all be dynamically tuned to match the exact overhead introduced by the specific recovery pattern.
>
> Therefore, while our prototype targets a commonly used configuration, we believe **MeCeFO naturally extends to other parallel strategies**, and exploring such extensions is a valuable direction for future work.
>
> ---
> > **Weakness 2. The authors haven't explored full system-level optimizations. Could they discuss the implications of this limitation?**
>
> While our current work has **demonstrated the algorithmic effectiveness of MeCeFO**, we have also **implemented a practical system prototype** with **measured throughput under fault scenarios**, showing that MeCeFO can be deployed and used in real training settings.
>
> However, this is not yet a full system-level implementation. In particular, **our fault scenarios are simulated rather than triggered by controllable hardware-level failures**. Achieving a complete system integration—where failures are **automatically detected, injected, and handled at the infrastructure level**—would require **low-level access to the cluster scheduler or hardware stack**, which is beyond our current resource scope.
>
> In this sense, while our work covers both algorithm design and usable implementation, **a fully integrated system with real-time fault injection and recovery logic lies slightly beyond the scope of this paper**, and we consider it an important direction for future work.
>
> ---
> > **Question 1. MeCeFO claims to avoid reserving half the GPU memory. Could the authors provide more data on memory usage?**
>
> As discussed in **Section 3**, MeCeFO reduces memory usage primarily through two key techniques:
>
> - **Skip Connection for Fault Recovery:** By skipping the update of attention parameters during the recovery phase, we save both the **activation storage** and the **optimizer states** for these parameters. In standard Transformer blocks, the multi-head attention (MHA) module includes Q/K/V/O projections totaling roughly $4h^2$ parameters, while the feedforward network (FFN) has about $8h^2$. Thus, the MHA typically takes up around **1/3 of the total parameters per block**, and skipping their updates leads to significant memory savings during training.
> - **Selective Gradient Checkpointing:** We apply gradient checkpointing to **only the FFN module**, which avoids storing intermediate activations for FFN and instead recomputes them during backpropagation. This choice balances memory savings and recomputation overhead, and further **reduces activation memory** substantially.
>
> Together, these two techniques allow MeCeFO to avoid the need for **pre-reserving up to half of GPU memory** for recovery.
>
> We refer the reviewer to **Table 4** (reproduced below) for quantitative memory usage comparisons:
>
> |Method|Throughput/(Tokens$\cdot$s$^{-1}$) (bsz=256)|Memory/GB (bsz=256)|Throughput/(Tokens$\cdot$s$^{-1}$) (bsz=512)| Memory/GB (bsz=512)|
> |-|-|-|-|-|
> |MeCeFOmrl|19.11k|76.13|-|OOM|
> |MeCeFOrl|30.23k|54.65|-|OOM|
> |MeCeFOl|26.41k| 38.48|23.86k|70.71|
> |**MeCeFO**|28.06k|39.21|27.19k|73.25|
> |MeCeFO w/o Fault|28.12k|41.52|30.04k|76.05|
>
> **Interpretation**:
>
> * **MeCeFOmrl** represents the baseline with no memory-saving techniques.
> * **MeCeFOrl** applies only the skip connection, already reducing memory by \~22GB.
> * **MeCeFOl** adds **selective gradient checkpointing** on top of the skip connection, achieving even more memory reduction (from 76.13GB → 38.48GB) for batch size 256, and remains runnable at batch size 512.
> * **MeCeFO** combines all three techniques and keeps memory usage close to the fault-free baseline, thus **avoids the need to statically reserve half of the memory**, instead dynamically adapting with significantly **lower memory footprint** and maintaining high throughput.
>
> ---
> > **Question 2. How would even more frequent failures or slower recovery affect training?**
>
> Thank you for the insightful question. In general, the **training efficiency under failures is primarily determined by the expected fraction of failed nodes at any given time**. This fraction is governed by the **ratio between failure rate and recovery rate**:
>
> * If failures become **more frequent** or **recovery becomes slower**, the expected fraction of failed nodes increases. This leads to **larger degradation** in training performance for all methods.
> * Conversely, if failures are less frequent or recovery is faster, the average number of unavailable nodes at any time is lower, and **performance degradation is mitigated**.
>
> This relationship is also evident in our experiments: from *low-frequency* to *high-frequency* settings, the **failure-to-recovery ratio increases**, and we observe **increasing gaps between MeCeFO and other baselines** in terms of training throughput. This confirms that:
>
> - **All methods degrade** as the failure-to-recovery ratio increases.
> - **MeCeFO degrades much more gracefully**, maintaining significantly better training speed under harsher conditions.
>
> To further support this, we conducted an additional experiment using MeCeFO to pre-train LLaMA-1B under a failure frequency of once every 10 minutes and a recovery time of 40 minutes. This setting yielded a **validation perplexity of 15.81**, closely matching the **15.83** observed under the 30-minute failure, 2-hour recovery setting. These results reinforce our claim that **training performance is primarily influenced by the failure-to-recovery ratio**, rather than the absolute failure rate alone.
>
> In the **extreme case** where all nodes are permanently failed (e.g., recovery never happens and no new nodes are added), **no distributed algorithm can make progress**, including MeCeFO. However, within practical ranges of node unavailability, **MeCeFO exhibits strong robustness**, and its **relative advantage grows as fault conditions worsen**.
>
> ---
> > **Question 3. Could MeCeFO be applied to other training regimes (e.g., fine-tuning) or different model types?**
>
> We appreciate the suggestion. To assess the broader applicability of MeCeFO, we conducted two additional evaluations covering both fine-tuning and alternative model architectures.
>
> - **Fine-tuning on GLUE:** We fine-tuned LLaMA-1B-N/L/M/H checkpoints—originally pre-trained with MeCeFO under no-fault, low-, medium-, and high-frequency failure scenarios, respectively—on the GLUE benchmark under the same corresponding failure settings. As shown in Table 1, MeCeFO-trained checkpoints maintained strong performance across all tasks, confirming that the benefits of MeCeFO extend to fine-tuning:
>
> **Table 1: Fine-tuning results on the GLUE benchmark using MeCeFO-pretrained LLaMA-1B models**
>
> | |CoLA|STS-B|MRPC|RTE|SST2|MNLI|QNLI|QQP|Avg|
> |-|-|-|-|-|-|-|-|-|-|
> |LLama-1B-N|46.93|**89.21**|**89.12**|62.61|**92.36**|**81.82**|88.61|89.83|80.06|
> |LLama-1B-L|46.86|89.14|88.92|62.59| 92.31|81.78|88.58|**90.07**|80.03|
> |LLama-1B-M|**47.21**|89.14|88.84|**63.18**|92.25|81.80|88.61|90.02|**80.13**|
> |LLama-1B-H|46.67|89.16|88.87|62.58|92.30|81.71|**88.66**|89.94|79.99|
>
> - **Support for other model architectures:** To evaluate MeCeFO’s compatibility with different model types, we implemented a DeepSeek V3-style architecture featuring Multi-Head Latent Attention (MLA) and a Mixture-of-Experts (MoE) design. This model has 1.2B total parameters, with 0.1B active per token. We pre-trained it under the same failure scenarios, and observed consistently robust performance, as illustrated in Table 2.
>
> **Table 2: Validation perplexities of MoE models trained with MeCeFO under various failure frequencies**
>
> |Model|No Fault|Low-frequency Fault|Medium-frequency Fault|High-frequency Fault|
> |-|-|-|-|-|
> |MLA+MoE|16.17|16.22|16.37|16.43|
>
> These results demonstrate that **MeCeFO remains effective beyond pre-training**, extending to both **fine-tuning tasks and structurally different models**.
>
> ---
> We thank the reviewer again for the careful comments and valuable suggestions. We hope these responses can clarify the reviewer's questions and are more than happy to address any further comments or questions.

---

> > ### Comment · Reviewer_16Gg · 2025-08-05
> >
> > Thank you for your detailed response.
> >
> > It has resolved all my concerns, and I have decided to maintain my rating score, and increase my confidence score to 4.

---

> > > ### Author Response · Authors · 2025-08-07
> > >
> > > Dear Reviewer 16Gg,
> > >
> > > Thank you for your positive feedback and for taking the time to review our work. We truly appreciate your support and are glad that our clarifications addressed your concerns.
> > >
> > > Best regards,
> > >
> > > Authors of Submission 15836

---

### Official Review · Reviewer_hbK1 · 2025-07-03

**Clarity:** 2
**Significance:** 3
**Originality:** 3
**Rating:** 5
**Confidence:** 3

**Summary:**

This paper proposes a new fault tolerance optimization technique for LLM training named MeCeFO. This approach effectively handle the failed node recomputation sceanrios by employing 1) selective skip reconnection 2) activation recomputation 3) low rank gradient approximation to reduce the computational and memory overhead when neighbour is need to recompute the result for failed node. The paper supports this approach by providing convergence analysis and empirical evaluation.

**Questions:**

- In line 236, I believe the author justified assumption 3 through empirical observation only on the error rate. However, the empirical observation is only based on a simple observation of llama 1B pre-training. Could the author elaborate a bit more on whether this observation is enough / further ablation on a larger model / more setting is required to justify this assumption fully?
- By choosing the NDB strategy among the DP group, the author assumed that the failure domain should be constrained within one DP group, ensuring that there would always be a node in the same DP group to recover the training progress. I wonder what would be the case if the failure domain is the entire DP group (i.e., if one switch corresponding to that DP group failed). Could the author discuss the impact of the failure domain in more detail?
- The evaluation and discussion primarily focused on llama-like dense architecture. I am curious about whether these techniques can be applied to MoE models, and if so, whether the sparsity in MoE models can facilitate additional resource savings and recomputation. Additionally, I am interested in determining the optimal placement of the router computation within the failure domain. Also, if the skip connection is suitable for MHA only or is it good on MLA, etc.

**Ethical Concerns:**

["NO or VERY MINOR ethics concerns only"]

**Final Justification:**

The author has resolved most of my concerns regarding the paper. With the recommended modification that I suggested, I am satisfied to give the score (5) on the paper.

**Limitations:**

The author adequately addresses most of the work's limitations.

**Quality:**

3

**Strengths And Weaknesses:**

strength

- Novelty. The author presents a novel method for resource-efficient recomputation in fault-tolerant training, based on the observation of the impact of placing skip connections at the MHA component and utilizing approximation techniques to reduce gradient computation complexity. This approximation technique is novel for weighing the tradeoff between accuracy and computational complexity.
- Robust Evaluation and Convergence Analysis. The paper provides a robust evaluation with different failure settings and various model size of pre-training to demonstrate the effectiveness of their approach. They also provide a convergence analysis for the proposed approach, showing that it achieves convergence rates comparable to those of SGD.
- Well-motivated optimization. The paper’s selection of techniques to minimize computational and memory overhead is well-reasoned and provides a good middle ground for the fault-tolerant theme they identified - approximation is sometimes good enough, and exact retracing of computation might not be necessary. I believe that in the paper, the approximation theme is well-motivated and effectively utilized.

weakness

- The author discusses hybrid parallelism, focusing on addressing the failure node through DP group recomputation. However, the TP group is likely the ‘minimal group for nodes’ under the training strategy, i.e., nodes → TP group → PP group → DP group. In this case, a failed node would be present in a TP group. If I understand the proposed approach correctly, the entire TP group / PP group needs to be recomputed as they would be counted as one ‘DP shard’ of the DP group, which results in recomputation of ‘healthy node’. Could the author discuss what would happen in this case, specifically whether the healthy node needs recomputation or can cache the result, and whether their approach could be directly applied to the TP group?
- In the empirical evaluation, the node recovery time seems to be a bit confusing - could the author explain the rationale for choosing node recovery time that is up to an hour scale, and the recovery time differs in each setting (rather than, for example, a constant recovery time of say, 2h). In addition, based on previous literature, I wonder if the low frequency setting reporting in the evaluation is not representative, and shorter failure period results (i.e., 5 minutes, 15 minutes failure interval) should also be considered.
- In practice, different failure causes likely result in different failure recovery times and frequencies - some might simply be a flip of a network connection, whereas others require hardware replacement. The uniform failure rate/failure mode assumed in the evaluation is an idealized situation considering what would happen at a real, large-scale LLM pre-training setting. In addition, sometimes due to the failure cause(e.g, failed network switch), more than one node would fail with a locality. Could the author discuss how their approach would be adapted to these various failure modes and failure rates, and how the worst-case and/best-case scenarios may affect this approach?

---

> ### Author Rebuttal · Authors · 2025-07-31
>
> We thank the reviewer for acknowledging our novelty, theoretical and experimental results, as well as the detailed comments and suggestions. All questions have been clarified as best as we can, and we are glad to address any further comments or questions.
>
> ---
> > **Weakness 1. While this work focuses on DP+PP parallelism, TP group is likely the minimal group for nodes is real settings. Could the author discuss what would happen in the TP setting?**
>
> While our main discussions focus on the DP+PP setting, we emphasize that **all three core mechanisms of MeCeFO—skip connections, selective activation recomputation, and low-rank gradient approximation—are applied locally at the individual node level**, and thus MeCeFO can **naturally extend to TP scenarios**.
>
> In the presence of a failed node within a TP group, a practical adaptation is to **redistribute its workload across sibling TP ranks within the same PP stage**. This avoids recomputing the entire TP group and instead only requires additional computation on a small subset of neighboring nodes. When TP > 2, this redistribution introduces **less than 1× overhead per node**, which allows us to adopt a more **conservative fallback strategy** to control the error.
>
> Moreover, MeCeFO is inherently **flexible and tunable**. Although in the standard configuration, our techniques reduce approximately half the workload per fallback, they can be easily adjusted to align with the parallelism structure and accuracy-performance tradeoff in TP settings:
>
> * **Skip connections** can selectively bypass only a fraction of sub-modules,
> * **Gradient checkpointing** can retain more activations and reduce recomputation depth,
> * **Low-rank gradient approximation** can adopt higher ranks for improved fidelity.
>
> ---
> > **Weakness 2. Why does the node recovery time differ in each setting? In addition, evaluation on shorter failure period (i.e., 5/15 minutes) are expected.**
>
> In our setup, the relatively **short failure intervals and variable recovery durations per node** serve as a mechanism to **simulate the aggregate failure dynamics of large-scale distributed systems using a small-scale cluster**. As discussed in [1] (section 7.2), such an experimental setting can cover a wide spectrum of environments.  Therefore, our increasing the per-node failure frequency is to simulate a larger distributed system where node failures happen more frequently, within a fixed number of hardware resources, rather than assuming every single-node is easier to fail.
>
> Accordingly, the choice of recovery time follows a similar rationale. In real systems, the failure and recovery rates are typically linked through shared infrastructure and operational constraints. Hence, we design the **recovery-to-failure rate ratio** to reflect this relationship, with recovery rates increasing **sublinearly** relative to failure rates. This models realistic scenarios in ultra-large clusters, where repair and maintenance resources may become bottlenecks under high failure pressure.
>
> We argue that it is this **ratio between failure and recovery rates**—rather than their absolute values—that is most critical, as it determines the **steady-state proportion of healthy nodes** and thus the overall system behavior and algorithm robustness. To further support this point, we include an additional experiment, where failures occur **once every 10 minutes**, and recovery takes **40 minutes**. Under this setting, we pre-trained the LLaMA-1B model using MeCeFO and obtained a **validation perplexity of 15.81**, which is consistent with the performance (15.83) observed in our high-frequency corruption experiments.
>
> > [1] Jang, I., Yang, Z., Zhang, Z., Jin, X., & Chowdhury, M. (2023, October). Oobleck: Resilient distributed training of large models using pipeline templates. In Proceedings of the 29th Symposium on Operating Systems Principles (pp. 382-395).
>
> ---
> > **Weakness 3 (Question 2). In practice, different failure causes likely result in different failure recovery times and frequencies. The uniform failure rate/failure mode is an idealized situation. In addition, failures like failed network switch may cause more than one node to fail. Could the authors discuss how MeCeFO would be adapted to these scenarios?**
>
> In this work, MeCeFO is **explicitly scoped to address one of the most common case: single-node failures**, which, according to large-scale system studies [2], account for approximately **82.5%** of observed failure events in production-scale LLM training clusters. This single-node failures setting is the standard practice [1,3] in the field of fault-tolerant training.
>
> We acknowledge that more complex scenarios, such as **localized multi-node failures**, are indeed important in real-world systems. While MeCeFO is specifically designed to handle **single-node failures**, which represent the majority of observed failures, it can be naturally integrated into a **hierarchical robustness framework**—serving as the **node-level component** within a broader fault-tolerance strategy.
>
> In summary, while MeCeFO currently targets single-node failure resilience, it is **designed to be modular and composable**, making it compatible with broader strategies for tolerating diverse failure types.
>
> > [2] Dong, J., Luo, B., Zhang, J., Zhang, P., Feng, F., Zhu, Y., ... & Fu, B. (2025, March). Enhancing Large-Scale AI Training Efficiency: The C4 Solution for Real-Time Anomaly Detection and Communication Optimization. In *2025 IEEE International Symposium on High Performance Computer Architecture (HPCA)* (pp. 1246–1258). IEEE.\
> > [3] Thorpe J, Zhao P, Eyolfson J, et al. Bamboo: Making preemptible instances resilient for affordable training of large DNNs[C]. 20th USENIX Symposium on Networked Systems Design and Implementation (NSDI 23). 2023: 497-513.
>
> ---
> > **Question 1. The empirical justification of Assumption 3 is only based on pre-training LLaMA 1B. Further ablation on a larger model / more setting is expected.**
>
> Regarding the scalability of this assumption, we offer the following clarifications:
> - **Model size and depth**: It is true that deeper models may exhibit longer error propagation paths. To assess the generality of our assumption beyond the 1B case, we conducted **additional experiments** on a **7B model with 32 transformer layers**, constrained by our current computational resources. The results shown in Table 1 and 2 below reveal similar trends in gradient approximation error, suggesting that Assumption 3 remains valid at larger scales.
>
> Table 1: The single-batch relative error of pre-training LLaMA-7B on the C4 dataset under high-frequency failure scenarios. (Corresponds to Figure 4)
>
> |Relative Error Interval|[0.000,0.050)|[0.050,0.100)|[0.100,0.150)|[0.150,0.200)|[0.200,0.250)|[0.250,0.300)|[0.300,0.350)|[0.350,0.400)|[0.400,0.450)|
> |-|-|-|-|-|-|-|-|-|-|
> |Density|3.87|2.27|4.33|2.67|2.87|1.67|1.67|0.53|0.13|
>
> Table 2: The full-batch relative error of pre-training LLaMA-7B on the C4 dataset under high-frequency failure scenarios. (Corresponds to Figure 5)
>
> |Relative Error Interval|[0.000,0.025)|[0.025,0.050)|[0.050,0.075)|[0.075,0.100)|[0.100,0.125)|[0.125,0.150)|[0.150,0.175)|[0.175,0.200)|[0.200,0.225)|
> |-|-|-|-|-|-|-|-|-|-|
> |Density| 10.13|6.00|3.47|6.27|8.80|0.27|1.60|2.93 |0.53|
>
> - **Data parallel (DP) averaging effect**: In practice, hardware failures are often sparse. Due to **gradient averaging across DP ranks**, the impact of a single node’s gradient deviation is diluted. As the number of DP replicas increases, the overall effect of any one faulty node diminishes. This effect enhances the robustness of our method in **larger-scale deployments**, where partial failures are more frequent, but individually less impactful.
>
> Taken together, while more comprehensive validation under a broader set of model scales and failure modes remains a valuable direction for future work, our current empirical evidence and theoretical reasoning suggest that Assumption 3 holds **under realistic and practically relevant conditions**.
>
> ---
> > **Question 3. The evaluation is limited to LLaMA-like dense architecture. Evaluation on MoE models and MLA is expected. It is interesting whether MoE's sparsity can facilitate resource saving and how to determine the router computation upon failure.**
>
> We sincerely thank the reviewer for this insightful and forward-looking question.
>
> - **Applicability to MoE:** MoE models share a modular structure similar to dense models, consisting of feedforward blocks and attention layers. Despite sparsity at the expert-selection level, **under balanced routing, each node remains significantly utilized**, meaning MeCeFO’s memory and computation reduction strategies—such as skip connections and low-rank approximations—can still provide benefits. In cases of **load imbalance**, where some experts are underutilized, there might be further opportunity to reduce fallback aggressiveness, potentially improving accuracy.
> - **Router computation and failure domains:** Routing computations in MoE generally occur during the forward pass and are relatively inexpensive compared to expert MLP computations. We therefore anticipate that **router computations could remain unchanged during fallback**, as MeCeFO already preserves the forward path.
> - **Application to MLA:** We have experimented with a Deepseek V3-style model incorporating Multi-Head Latent Attention and Mixture-of-Experts, totaling 1.2B parameters with 0.1B active parameters. This model was trained under various fault scenarios—no fault, low-frequency, medium-frequency, and high-frequency faults—with corresponding validation perplexities as follows:
>
> |Model|No Fault|Low-frequency|Medium-frequency|High-frequency|
> |-|-|-|-|-|
> |MLA+MoE|16.17|16.22|16.37|16.43|
>
> ---
> We thank the reviewer again for the careful comments and valuable suggestions. We hope these responses can clarify the reviewer's questions and are more than happy to address any further comments or questions.

---

> > ### Comment · Reviewer_hbK1 · 2025-08-05
> >
> > I believe the author has answered most of my concerns, except for the part regarding the failure mode. In particular, I would ask the author to provide more detail on the recovery-to-failure rate ratio they had discussed to justify their experiment setting. Could the author elaborate on how the existing setting of their experiment corresponds to each of the failure modes mentioned earlier?

---

> ### Author Response · Authors · 2025-08-05
>
> Thank you for your thoughtful follow-up. We are happy to provide further clarification on how the **failure-to-recovery ratio** in our experiments corresponds to each experimental setting.
>
> In our experiments, although we use a **32-GPU cluster**, the failure/recovery events are **intentionally amplified** to simulate large-scale deployments with hundreds or thousands of nodes. Specifically:
>
> * In the **Low Frequency Failure** setting, our system experiences **1 node failure every 2 hours**, corresponding to a per-node failure rate of **$1/64$ failures per hour**. Since such a high per-node failure rate is unrealistic in practice (rarely does every node in a large-scale cluster fail once every 3 days), we interpret this as **each node simulating $N \gg 1$** real nodes, each with a much lower failure rate of **$1/(64N)$ per hour**. The recovery rate (1 recovery every 4 hours) corresponds to a per-node recovery rate of $1/(128N)$ per hour.
>
> * In the **Medium Frequency Failure** setting, the system sees **1 node failure per hour**, implying a per-node failure rate of **$1/32$ per hour**. Keeping the real-node failure rate fixed at **$1/(64N)$**, each simulated node corresponds to **$2N$** real nodes. To model larger clusters where recovery becomes more complex (due to increased scheduling and detection overhead), we adjust the recovery-to-failure ratio accordingly, resulting in a **recovery rate of $1/(192N)$** per node, equivalent to 1 global recovery every 3 hours.
>
> * Similarly, in the **High Frequency Failure** setting, the system sees **1 node failure every 30 minutes**, corresponding to each simulated node representing **$4N$** real nodes. We keep the real-node failure rate at **$1/(64N)$** per hour, but further **lower the recovery rate** to **$1/(256N)$** per node to emulate the compounding repair complexity in ultra-large clusters.
>
> The following table summarizes this mapping more clearly:
>
> | Setting                  | Simulated Cluster Size | Failure Frequency (per node per hour) | Recovery Frequency (per node per hour) | Equivalent Real Nodes per Simulated Node | Equivalent Failure Frequency (per real node per hour) | Equivalent Recovery Frequency (per real node per hour) |
> | ------------------------ | ---------------------- | ---------------------------- | ----------------------------- | ---------------------------------------- |-|-|
> | Low-Frequency Failure    | 32 GPUs                | $1/64$              | $1/128$                | $N$                                        | $1/(64N)$ | $1/(128N)$|
> | Medium-Frequency Failure | 32 GPUs                | $1/32$            | $1/96$                 | $2N$                                       | $1/(64N)$ | $1/(192N)$ |
> | High-Frequency Failure   | 32 GPUs                | $1/16$               | $1/64$                 | $4N$                                       | $1/(64N)$ | $1/(256N)$ |
>
> We hope this detailed explanation clarifies how our settings correspond to realistic failure patterns across different cluster scales. Our intention was to **capture realistic node dynamics** by adjusting per-node rates to simulate system-wide behavior while respecting the complexity of large-scale fault recovery (the larger the scale, the lower the recovery-to-failure ratio).
>
> Please let us know if further clarification is helpful—we greatly appreciate your continued engagement.

---

> > ### Comment · Reviewer_hbK1 · 2025-08-05
> >
> > Thank you for further explaining the settings on the failure-recovery rate.
> > I believe that your table here has resolved my concern in the previous comment, and I recommend that the authors consider including this in the main body of the paper to help explain the concept better.
> >
> > In addition, I also suggest that the authors include a discussion on how their single-node fault-tolerance strategy can be composed with other fault-tolerance strategies to handle reliability issues in a more realistic setting (as discussed with localized multi-node failure in the comment). This would help to further understand the scope and capabilities of the work in a broader context.
> >
> > With that, I have updated my score to reflect of my new assesment of the work.

---

> > > ### Author Response · Authors · 2025-08-06
> > >
> > > Thank you for your suggestions and constructive discussion. We will incorporate the clarified failure-recovery mapping and the associated table into the revised version of the paper to better convey the experimental design and assumptions.
> > >
> > > In addition, we will include the following discussion in the revised paper:
> > >
> > > > MeCeFO serves as a complementary component to system-level fault-tolerant frameworks, enabling the construction of more robust and resilient training systems. While we do not specifically endorse any particular solution, we provide the following as illustrative examples to demonstrate the feasibility and flexibility of integrating MeCeFO into a broader system:
> > > > - **MeCeFO** efficiently handles isolated node failures.
> > > > - **[1]** proposes a hierarchical detection mechanism that targets switch-level and interconnect failures in distributed networks.
> > > > - **[2]** addresses a wider range of issues including node freezing, communication disruptions, and software errors encountered during large-scale LLM training.
> > >
> > > > By integrating MeCeFO with these diverse system-level strategies, we can produce a more comprehensive and reliable fault-tolerant training system.
> > >
> > > We hope this integrated perspective will help clarify the scope and practical deployment potential of MeCeFO in realistic and large-scale environments.
> > >
> > > Thank you again for your valuable feedback and for updating your score based on the revised assessment.
> > >
> > > > [1] Gupta, B. K., Mundra, A., & Rakesh, N. (2014). Failure detection and recovery in hierarchical network Using FTN approach. arXiv preprint arXiv:1401.8131.\
> > > > [2] Wu, B., Xia, L., Li, Q., Li, K., Chen, X., Guo, Y., ... & Li, S. (2023). Transom: An efficient fault-tolerant system for training llms. arXiv preprint arXiv:2310.10046.

---

### Official Review · Reviewer_cmnw · 2025-07-20

**Clarity:** 3
**Significance:** 3
**Originality:** 4
**Rating:** 4
**Confidence:** 4

**Summary:**

This paper introduces MeCeFO, a novel fault-tolerant optimization algorithm designed to ensure robust training with minimal overhead. When a computing node fails, MeCeFO efficiently transfers the workload to a neighboring node while applying memory- and computation-saving techniques. It incorporates three key algorithmic innovations: skip-connection to approximate attention during backpropagation, recomputation to reduce activation memory, and low-rank gradient approximation for efficient gradient estimation. Theoretically, MeCeFO achieves convergence comparable to conventional distributed training, while empirically demonstrating strong resilience under frequent failure conditions.

**Questions:**

1. This paper states that "The convergence depends on the quality of final weights, not the precise trajectory." Can you validate it by experiments?

2. In real systems, failures tend to cluster on certain nodes (i.e., non-i.i.d.), causing the NDB to repeatedly fallback on the same “victim” nodes. Please analyze how this asymmetric load might induce training bias or failure.

3. Section 4 provides an upper-bound theoretical analysis on the convergence impact of skip/backward approximation. Can you provide worst-case analysis and explore whether these errors could be amplified during early training stages or critical layers?

**Ethical Concerns:**

["NO or VERY MINOR ethics concerns only"]

**Final Justification:**

I am satisfied with rebuttal.

**Limitations:**

Nil

**Paper Formatting Concerns:**

Nil

**Quality:**

3

**Strengths And Weaknesses:**

Strength:
1. This paper proposes MeCeFO, a novel fault-tolerant optimization algorithm that enhances efficiency in distributed training.
2. This paper provides theoretical guarantees showing MeCeFO achieves convergence rates comparable to standard distributed SGD under mild assumptions.
3. This paper demonstrates MeCeFO’s superior resilience to frequent failures, maintaining high training throughput and robust model performance.


Weakness:
1. This paper states that "The convergence depends on the quality of final weights, not the precise trajectory," which is not validated by the experiments. The experiments verify "the quality of final weights" only through final perplexity, without evaluating downstream transfer capabilities (e.g., zero-shot or few-shot task performance), training stability, or generative quality metrics (such as BLEU).

2. A key implicit assumption behind the MeCeFO mechanism is that failures are sparse and “uniformly random,” with roughly equal failure probabilities across nodes and the ability for neighbors to rotate load. In real systems, failures tend to cluster on certain nodes (i.e., non-i.i.d.), causing the NDB to repeatedly fallback on the same “victim” nodes. The paper does not analyze how this asymmetric load might induce training bias or failure. This raises concerns about the robustness and generalizability of the proposed method.

3. Section 4 provides an upper-bound theoretical analysis on the convergence impact of skip/backward approximation, but it lacks worst-case analysis and does not explore whether these errors could be amplified during early training stages or critical layers. For fault tolerance mechanisms, a key question is whether the "worst-case deviation is controllable." Currently, the theoretical discussion focuses mainly on average-case scenarios, which cannot guarantee system stability.

---

> ### Author Rebuttal · Authors · 2025-07-31
>
> We thank the reviewer for acknowledging our theoretical and experimental results, as well as the detailed comments and suggestions. All questions have been clarified as best as we can, and we are glad to address any further comments or questions.
>
> ---
> > **Weakness 1 (Question 1). The paper states that convergence depends on the quality of final weights rather than precise trajectory. Evaluation regarding downstream transfer capabilities, training stability or generative quality metrics are expected.**
>
> We appreciate the reviewer’s concern and acknowledge that our statement — “The convergence depends on the quality of final weights, not the precise trajectory” — may have caused some confusion. Our intended message is that, for the goal of training a performant model, the specific optimization path is less critical as long as the final checkpoint achieves strong performance. In this context, our proposed training modifications are justified as long as they lead to a model of comparable or better quality.
>
> We agree that final validation perplexity alone may not fully capture the quality of a model, and that broader evaluation—including training stability, downstream task performance, and generative quality—is important. Regarding training stability, we did not observe any issues such as gradient spikes or divergence in either MeCeFO or standard training, though a more systematic study of robustness is a valuable future direction.
>
> For downstream evaluation, which is more straightforward to assess, we have conducted additional experiments. The following zero-shot evaluation results on BoolQ (Nautral Language Understanding), ARC-Easy (Reasoning), PIQA (Commensense Understanding) and TruthfulQA (Truthfulness) show that LLaMA-1B-N/L/M/H (LLaMA-1B trained with MeCeFO under no-fault/low-frequency/medium-frequency/high-frequency fault conditions) have similar performance:
>
> **Table 1: Zero-shot evaluation results using MeCeFO-pretrained LLaMA-1B models**
> | Model   | BoolQ     | ARC-Easy  | PIQA      | TruthfulQA-MC2 | Avg.      |
> | ---------- | --------- | --------- | --------- | -------------- | --------- |
> | LLama-1B-N | 0.579     | **0.459** | 0.682     | 0.427          | 0.537     |
> | LLama-1B-L | **0.594** | 0.455     | 0.674     | **0.451**      | **0.544** |
> | LLama-1B-M | 0.571     | 0.446     | 0.678     | 0.425          | 0.530     |
> | LLama-1B-H | 0.587     | 0.454     | **0.684** | 0.417          | 0.536     |
>
> We also fine-tuned LLaMA-1B-N/L/M/H checkpoints—originally pre-trained with MeCeFO under no-fault, low-, medium-, and high-frequency failure scenarios, respectively—on the GLUE benchmark under the same corresponding failure settings. As shown in Table 2, MeCeFO-trained checkpoints maintained strong performance across all tasks, confirming that the benefits of MeCeFO extend to fine-tuning:
>
> **Table 2: Fine-tuning results on the GLUE benchmark using MeCeFO-pretrained LLaMA-1B models**
>
> |            | CoLA      | STS-B     | MRPC      | RTE       | SST2      | MNLI      | QNLI      | QQP       | Avg       |
> | ---------- | --------- | --------- | --------- | --------- | --------- | --------- | --------- | --------- | --------- |
> | LLama-1B-N | 46.93     | **89.21** | **89.12** | 62.61     | **92.36** | **81.82** | 88.61     | 89.83     | 80.06     |
> | LLama-1B-L | 46.86     | 89.14     | 88.92     | 62.59     | 92.31     | 81.78     | 88.58     | **90.07** | 80.03     |
> | LLama-1B-M | **47.21** | 89.14     | 88.84     | **63.18** | 92.25     | 81.80     | 88.61     | 90.02     | **80.13** |
> | LLama-1B-H | 46.67     | 89.16     | 88.87     | 62.58     | 92.30     | 81.71     | **88.66** | 89.94     | 79.99     |
>
> Results show that models trained with MeCeFO during both pre-training and fine-tuning achieve downstream task accuracies comparable to those trained with standard methods, supporting the claim that our approach does not compromise generalization or transfer ability.
>
> ---
> > **Weakness 2 (Question 2). MeCeFO mainly considers sparse and "uniformly random" failures. In real systems, failures tend to cluster on certain nodes, implying asymmetric load, raising concerns about MeCeFO's robustness and generalizability.**
>
> We thank the reviewer for raising this important concern. We agree that real-world failures can exhibit asymmetric and non-i.i.d. patterns, and understanding how MeCeFO performs under such conditions is crucial. While our current design assumes uniformly random failures for simplicity, we argue that MeCeFO remains robust even when this assumption is relaxed. Below, we provide both theoretical reasoning and supporting experimental evidence.
>
> Specifically, we consider two illustrative failure patterns across two data-parallel pipelines, $P_1: N_{11} \rightarrow N_{12}$ and $P_2: N_{21} \rightarrow N_{22}$, where $N_{ij}$ denotes the node in $P_{i}$'s $j$-th pipeline stage.
> (a) failures alternate between $N_{11}$ and $N_{21}$, causing NDB to fallback alternately to $N_{12}$ and $N_{22}$;
> (b) $N_{11}$ always fails while $N_{21}$ remains healthy, resulting in repeated fallback to $N_{12}$ only.
>
> In both cases, during even iterations (e.g., $t = 0, 2, 4, \dots$), the fallback and non-fallback pipelines are consistent. During odd iterations, although the fallback targets differ, each step still involves one fallback and one normal pipeline. Assuming the training data is evenly and randomly distributed across DP ranks, the model continues to encounter similarly distributed data across both fallback and normal paths over time. As a result, no significant training bias is introduced, and the overall training dynamics remain stable.
>
> To validate this analysis, we conducted an additional ablation study simulating persistent, non-uniform failure. Specifically, we randomly selected 5 GPUs and caused only these GPUs to fail throughout the entire training process, while others remained fully operational. All other experimental settings were kept the same as in the main study. The validation perplexities under this skewed failure setting are shown below, and closely match the results under uniform failure:
>
> | Setting  | No Fault | Low-frequency Fault | Medium-frequency Fault | High-frequency Fault |
> | -------- | -------- | ------------------- | ---------------------- | -------------------- |
> | LLaMA-1B (Asymmetric failure) | 15.49    | 15.54               | 15.62                  | 15.75                |
> | LLaMA-1B (Symmetric failure) | 15.49 | 15.51 | 15.61 | 15.83 |
>
> These results suggest that even in the presence of static and localized failures, MeCeFO maintains strong robustness and does not suffer from significant degradation in performance.
>
> That said, we acknowledge that a more systematic evaluation of highly skewed or clustered failure patterns is a valuable direction for future work, and we appreciate the reviewer for highlighting this important aspect.
>
> ---
> > **Weakness 3 (Question 3). The upper-bound theoretical analysis in section 4 focuses mainly on average-case rather than worst-case and does not explore whether these errors could be amplified during early training stages or critical layers.**
>
> We appreciate the reviewer’s concern regarding the potential amplification of gradient approximation errors, particularly during early training stages or in critical layers. This is indeed an important aspect of fault-tolerant training.
>
> Our theoretical analysis explicitly addresses this issue through **Assumption 3**, which states that at each iteration $t$, the approximate gradient computed by MeCeFO for weights $w^{(t)}$ must maintain a bounded relative error—no worse than a multiplicative factor of $(1 - \delta)$—compared to the true gradient from standard backpropagation. This assumption applies both to the stochastic and deterministic gradients.
>
> Importantly, our main convergence results—Theorem 1 and Corollary 1—hold even in the **worst-case scenario** where the relative error reaches this upper bound $(1 - \delta)$ at every iteration. In other words, the theoretical guarantees are **not merely average-case**, but are designed to ensure convergence under worst-case bounded degradation, as long as Assumption 3 holds.
>
> To support this assumption empirically, we provide error analyses in Figures 4 and 5, which show that in practice, including scenarios where the errors could occur in early training stages or critical layers, the observed gradient deviations stay well within the required bound. This suggests that even under stress conditions, MeCeFO maintains stability and robustness throughout training.
>
> ---
> We thank the reviewer again for the careful comments and valuable suggestions. We hope these responses can clarify the reviewer's questions and are more than happy to address any further comments or questions.

---

> > ### Author Response · Authors · 2025-08-07
> >
> > Dear Reviewer cmnw,
> >
> > Thank you again for your thoughtful feedback and positive evaluation of our submission. We are not sure whether our rebuttal has fully addressed your concerns, and would greatly appreciate it if you could let us know in case there are any remaining questions or points you'd like us to clarify.
> >
> > We’re happy to further discuss and sincerely appreciate your time and support.
> >
> > Best regards,
> >
> > Authors of Submission 15836

---

> > > ### Comment · Reviewer_cmnw · 2025-08-07
> > >
> > > I am satisfied with the rebuttal and remain positive with the paper.

---

> > > > ### Author Response · Authors · 2025-08-08
> > > >
> > > > Dear Reviewer cmnw,
> > > >
> > > > Thank you for your positive feedback and for taking the time to review our work. We truly appreciate your support and are glad that our clarifications addressed your concerns.
> > > >
> > > > Best regards,
> > > >
> > > > Authors of Submission 15836

---

### Comment · Area_Chair_5dVf · 2025-08-02
**Please Engage with Authors’ Responses During Rebuttal**

Dear Reviewers,

As we approach the rebuttal phase, I want to take a moment to remind you of the importance of carefully reviewing the authors’ responses. The rebuttal process is a valuable opportunity for authors to clarify misunderstandings, address concerns, and provide additional evidence or analysis to support their work.

When reviewing the authors’ responses, please:

1. Read their replies thoroughly to ensure you understand their points.
2. Assess whether your concerns have been adequately addressed.

Your engagement in this process helps ensure a fair and constructive review outcome. Thank you for your time and dedication to maintaining the high standards of NeurIPS. Your thoughtful participation in this phase is greatly appreciated!

Best regards,

Area Chair

---

### Note · Authors · 2025-08-12

We sincerely thank all reviewers for their active engagement, thoughtful feedback, and constructive suggestions.

The reviewers acknowledged our strengths:

1.  **Novel Algorithm Design:** We proposed MeCeFO, a fault-tolerant optimization algorithm designed for robust training with minimal overhead. It cleverly shifts workloads from failed nodes to their neighbors.
2.  **Key Technical Innovations:** MeCeFO incorporates three core innovations to maintain efficiency: skip-connections to approximate attention, activation recomputation, and low-rank gradient approximation.
3.  **Strong Theoretical and Empirical Results:** We proved that MeCeFO achieves a convergence rate comparable to traditional distributed training. Empirically, it demonstrates throughput on par with the failure-free baseline, even in environments with frequent node failures.

During the rebuttal phase, we:

1.  **Extended Experimental Validations:** We performed additional zero-shot and fine-tuning evaluations to further validate MeCeFO’s effectiveness.
2.  **Verified Broader Applicability:** We demonstrated that our method is applicable to other model architectures, such as MLA and MoE, and explored its use in other parallel training settings, like Tensor Parallelism.
3.  **Addressed Failure Scenarios in Depth:** We provided a more detailed analysis of failure recovery rates and experimentally showed stable performance under even harsher conditions (e.g., node failures every 10 minutes). We also discussed its compatibility with system-level strategies for handling issues like network interruptions.
4.  **Clarified Theoretical Concerns:** We offered a clear and detailed explanation that successfully addressed the reviewers' questions regarding our convergence theory.

Overall, our additional experiments and the fruitful discussions with the reviewers have successfully addressed their concerns, and we will incorporate these results and insights into the revised manuscript.

---

### Decision · Program_Chairs · 2025-09-17

**Decision:**

Accept (poster)

**Comment:**

**Summary**:
The paper introduces MeCeFO, a novel fault-tolerant optimization technique for large language model (LLM) training, designed to handle node failures efficiently. It employs selective skip reconnection, activation recomputation, and low-rank gradient approximation to minimize computational and memory overhead during failure recovery. The paper provides theoretical convergence guarantees and empirical evaluations on LLAMA models (350M, 1B, and 7B), comparing MeCeFO against baselines like Bamboo and Oobleck.

**Strengths**:

1. Reviewers unanimously praise the novelty of MeCeFO, particularly its use of skip connections at the multi-head attention (MHA) component and low-rank gradient approximations to balance accuracy and computational efficiency. The approach creatively leverages the noise-robust nature of SGD, offering a fresh perspective on fault-tolerant training.
2. The paper provides a robust convergence analysis, demonstrating that MeCeFO achieves convergence rates comparable to standard distributed SGD under mild assumptions. This theoretical backing strengthens confidence in the algorithm’s reliability.
3. The experimental results are compelling, showing MeCeFO’s superior resilience to frequent failures and maintaining high training throughput across various model sizes and failure settings. The inclusion of ablation studies in the appendix further supports the robustness of the approach.
4. The optimization techniques are well-reasoned, effectively addressing the trade-off between computational overhead and fault tolerance. The approximation-based strategy is particularly noted for its practical relevance in large-scale training.

**Weaknesses**:

1. Reviewers highlight that the evaluation focuses primarily on final perplexity without assessing downstream transfer capabilities (e.g., zero-shot or few-shot performance) or generative quality metrics like BLEU. This raises concerns about the generalizability of the final weights.
2. The assumption of sparse, uniformly random failures is criticized as unrealistic, as real-world failures often cluster on specific nodes, potentially leading to training bias or instability. The paper lacks analysis of how MeCeFO handles non-i.i.d. failure patterns or asymmetric loads.
3. While the convergence analysis is robust for average-case scenarios, it lacks worst-case analysis, particularly regarding error amplification in early training or critical layers. And it is questioned how the theoretical guarantee accounts for varying failure rates or statistical failure models.
4. MeCeFO is tailored to specific hybrid parallel strategies (e.g., DP group recomputation), but its applicability to tensor parallelism (TP) groups is unclear. Reviewers note that recomputation of entire TP/PP groups may involve healthy nodes, potentially increasing overhead.
5. The empirical evaluation assumes idealized failure models (constant frequency) and lacks discussion of varied failure causes, such as network flips or hardware replacements, which could affect recovery times and frequencies. The choice of node recovery times (up to an hour) is not well-justified, and shorter failure intervals (e.g., 5–15 minutes) are not explored.

**After Rebuttal,** all reviewers give positive scores of this paper and most of concerns are addressed. Thus, I recommend Accept.

However, the paper is not recommended for a spotlight or oral presentation due to significant concerns about the scope of its evaluation and assumptions. The algorithm modifies the training process substantially, yet the experiments lack large-scale validation to confirm its effectiveness in real-world, diverse failure scenarios. The absence of downstream task evaluations and the reliance on idealized failure models limit confidence in its generalizability. Additionally, the lack of worst-case theoretical analysis and clarity on handling varied parallelism strategies (e.g., TP groups) suggests that further investigation is needed to establish its robustness across practical settings.